# Hardware-Friendly Post-Training Quantization: Input- and Output-Channelwise Scale and Offset

## Abstract

Post-training quantization enables swift quantization of neural networks using a minimal calibration dataset. Prior approaches to low-bit quantization that use non-linear mixed-precision methods or different quantization bit allocations for each layer are not well-suited for hardware acceleration. Specifically, these methods tend to underperform dramatically on hardware with fixed integer bit width, particularly in extremely low-bit quantization scenarios. In response, we introduce an optimization-based method for uniform channel-wise quantization compatible with existing hardware. This approach does not increase memory requirements and results in only a marginal increase in computation. This strategy involves applying a specific multiplier to the result of the weighted activation products, yielding a more accurate result for the multiply-accumulate (MAC) operation in convolutional or fully-connected layers. We also present an optimization technique to determine the optimal channel grouping approach. We conducted tests on various CNN-based models to affirm the superiority of our proposed quantization scheme. Our proposed approach enhances accuracy in 2/4-bit weight and feature quantization by 1-5%p while only increasing the number of integer operations in convolutional-based networks by less than 1.5%.

## 1 Introduction

Among various model compression techniques (Han et al., 2015; Dong et al., 2017; Dong & Yang, 2019; Gou et al., 2021), quantization reduces the precision of the weights and activations which can significantly reduce the memory requirements and computational cost of a model. Quantization can be broadly categorized into uniform and non-uniform quantization (Choi et al., 2016; Cai et al., 2017). The former uses a fixed step size to represent the values, while the latter adapts the step size based on the distribution of the weight or activation values. Typically, (output) channel-wise uniform quantization is employed for its hardware-friendly nature (Jacob et al., 2018; Nagel et al., 2021), ease of fusing with batch normalization, and reduced high-cost floating-point operations by having different quantization scales for each output channel.

From a training perspective, there are two primary approaches of quantization in deep learning: Quantization Aware Training (QAT) (Xu et al., 2018; Esser et al., 2019; Kim et al., 2020) and Post-Training Quantization (PTQ) (Stock et al., 2019; Nagel et al., 2020; Li et al., 2021; Jeon et al., 2022). QAT trains a model with quantized weights and activations, which allows the model to adapt to the quantization process during training. In contrast, PTQ applies quantization to a pre-trained model, resulting in faster model compression but potentially causing a more significant loss in performance. Recent studies (Nagel et al., 2020; Li et al., 2021; Wei et al., 2022) have shown promising results by minimizing the difference between the activation mean of the full precision model and the quantized model (i.e. reconstruction loss). To optimize the reconstruction loss, these approaches determine whether to round up or down after per-weight-based quantization. Due to the distributional discrepancy between the full precision weights and their quantized counterpart, this rounding scheme is insufficient for fully optimizing the reconstruction loss in extremely low-bit quantization.

To address this, we introduce new parameters during PTQ that scale and offset the input and output channels to minimize the distributional gap. However, the input-channel scaling parameters intro-

duce computational overhead at inference in its naive form. Through grouping the input channels and utilizing the shift operation, we can absorb these parameters into the quantized model with a minimal computation overhead. Using a limited dataset (PTQ), we effectively optimized the layer through input-channel-wise grouping and verified that performance can be enhanced solely by employing shift operations, ensuring that all calculations remain within the integer domain. Our main contributions are as follows:

1. We propose a method that compensates for the discrepancies in full precision and quantized distributions that occur during the quantization of the product of weights and activations. This approach utilizes straightforward yet effective channel-wise scaling and offset parameters to tackle these disparities.

2. Our proposed method demonstrates its applicability in channel-wise quantization without requiring additional memory for parameters, with only less than 1.5% computation overhead in most networks during inference.

3. We show that our approach could be integrated with existing block reconstruction methods (Li et al., 2021; Wei et al., 2022; Zheng et al., 2022), further improving their performance.

## 2 PRELIMINARIES

**Notations** Scalars or vectors are represented in lowercase, while matrices and tensors are denoted in uppercase. For a real variable $a$, $\bar{a}$ and $\hat{a}$ signify the integer coded bit and quantized value, respectively. The quantized value $\hat{a}$ is the reconstructed approximation of the original variable $a$, i.e., $\hat{a} \approx a$. Parenthesized superscript $a^{(\cdot)}$ indicates the layer index. Among superscripts, $x$, $w$, $y$, and $z$ represent values for the input feature, weight, accumulated output feature, and output feature after batch normalization, activation, and re-quantization, respectively. Subscripts are used for indicating an element of a vector or matrix.

### 2.1 UNIFORM CHANNEL-WISE QUANTIZATION

Uniform channel-wise quantization (Jacob et al., 2018; Zhang et al., 2018; Wu et al., 2020; Nagel et al., 2021) apply a uniform quantization strategy separately to each output channel of a convolutional or fully-connected layer. This can be implemented using only integer operations. Given a real-valued $r$, the coded bit $\bar{r}$ and quantized value $\hat{r}$ can be expressed as [1]

$$\bar{r} = Q(r) = \texttt{clamp}(\lfloor r/s \rceil + z; min, max), \tag{1}$$

$$\hat{r} = (\bar{r} - z) \times s, \tag{2}$$

where the constants $s \in \mathbb{R}$ and $z \in \mathbb{Z}$ represent the scale and zero-point of quantization, respectively. $Q(\cdot)$ is the quantization function, $\lfloor \cdot \rceil$ denotes rounding to the nearest integer, and $\texttt{clamp}(v; a, b)$ constrains the value $v$ within the range $[a, b]$. Generally, quantization aims to make the approximation $\hat{r}$ as close as possible to the original value $r$.

Both the weight and activation must be appropriately quantized to operate a neural network on a quantized device. For a simpler hardware implementation with improved performance, typically, all weights corresponding to the same output channel share the same scale value with a zero-point of zero (symmetric quantization) (Wu et al., 2020), and all features share the same scale and zero-point (asymmetric quantization). When considering a convolution or fully-connected layer, where the weight matrix $W \in \mathbb{R}^{m \times c}$ is multiplied with the input feature $x \in \mathbb{R}^c$ to produce the output feature vector $y = Wx \in \mathbb{R}^m$, the $k$-th output feature $y_k$ can be computed as

$$y_k = \sum_i^c \hat{w}_{ki} \hat{x}_i = \sum_i^c (s_k^w \bar{w}_{ki}) \cdot (s^x(\bar{x}_i - z^x)) = s_k^w s^x \sum_i^c \bar{w}_{ki} \bar{x}_i - z^x s_k^w s^x \sum_i^c \bar{w}_{ki}, \tag{3}$$

where $s^w \in \mathbb{R}^m$ and $s^x \in \mathbb{R}$ are the scales of the weight and input feature, respectively. Note that the second term $z^x s_k^w s^x \sum_i^c \bar{w}_{ki}$ can be pre-calculated before inference time. Eq. (3) is for symmetric weight quantization and if this is not the case, i.e. $z^w \neq 0$, an additional multiplication and addition operation will be required.

---

[1] As in previous works (Nagel et al., 2021), we omit nonlinear activations such as ReLU in the equation.

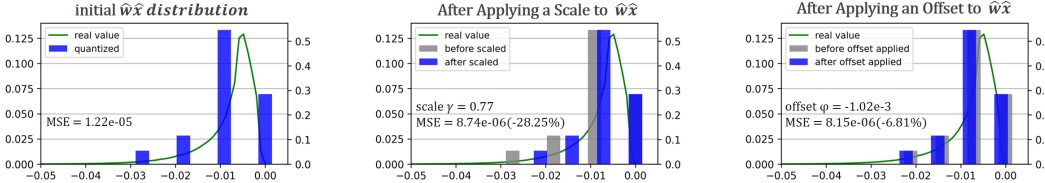

Figure 1: Fusion of convolutional layer and batch normalization, emphasizing the quantization process at different scales. The distinct colors represent the various scales of quantization.

Figure 2: Effect of the scale $\gamma$ and the offset $\varphi$ on the ResNet-18 layer1.conv1 layer in the 1/2-bit weigh/feature quantization example. MSE is the mean squared error between the full-precision product $wx$ and the quantized product $\hat{w}\hat{x}$.

Batch normalization (Ioffe & Szegedy, 2015), which typically follows convolution or fully-connected layers and operates as an affine function per output channel, can be performed in a single process and can be expressed in an affine form of $y_k = \alpha_k \sum_i^c \bar{w}_{ki} \bar{x}_i + \beta_k$ when combined with Eq. (3). The channel-wise parameters $\alpha_k \in \mathbb{R}^m$ and $\beta_k \in \mathbb{R}^m$ can be prepared before inference time, contributing to the computational simplicity – the primary reason of why per-output channel-wise uniform quantization is commonly utilized in hardware (Lin et al., 2016; Jacob et al., 2018; Krishnamoorthi, 2018; Nagel et al., 2021) (See Appendix A). Note that even without Batch normalization, re-quantization of $y_k$ is required, which carries out channel-wise floating point addition and multiplication due to the expanded bit-width of the output $y_k$ compared to those of $\bar{w}$ and $\bar{x}$ in Eq. (3). Fig. 1 illustrates the output feature map $Z$, which needs to be re-quantized to unify the scale values across all channels.

## 2.2 MEAN RECONSTRUCTION ERROR

Recent studies have shown significant improvements in post-training quantization performance (Li et al., 2021; Jeon et al., 2022; Wei et al., 2022; Zheng et al., 2022) even with a small calibration set by minimizing the mean reconstruction error of the intermediate feature maps. The loss increment caused by quantization can be analyzed by the second-order Taylor series expansion as follows:

$$\mathbb{E}_x[\Delta\mathcal{L}] = \mathbb{E}_x[\mathcal{L}(w + \Delta w)] - \mathbb{E}_x[\mathcal{L}(w)] \approx \Delta w^T \bar{g} + \frac{1}{2}\Delta w^T \bar{H} \Delta w. \tag{4}$$

Assuming that the gradient of the pre-trained model is nearly zero and the effect of higher order terms is negligible due to small $\Delta w$, the change in loss can be sufficiently approximated by the Hessian term, i.e., $\mathbb{E}_x[\Delta\mathcal{L}] \approx \frac{1}{2}\Delta w^T \bar{H} \Delta w$. Given a large number of parameters in deep learning models, directly calculating the Hessian matrix can be challenging and under some assumptions on Hessian and CNN architecture, it becomes

$$\Delta w^T \bar{H} \Delta w = \sum_l \Delta w^{(l)T} \bar{H}^{(l)} \Delta w^{(l)} = \sum_l ||\Delta W^{(l)} x^{(l-1)}||_2^2 = \sum_l ||\Delta y^{(l)}||_2^2. \tag{5}$$

Here, $\Delta w$ is the vectorized version of $\Delta W$ and $y^{(l)} = W^{(l)} x^{(l-1)}$ for the $(l-1)$-th layer's feature map $x^{(l-1)}$ and $l$-th layer's weight matrix $W^{(l)}$. Thus, minimizing the loss boils down to minimizing the block reconstruction error of each layer's output feature map (Li et al., 2021).

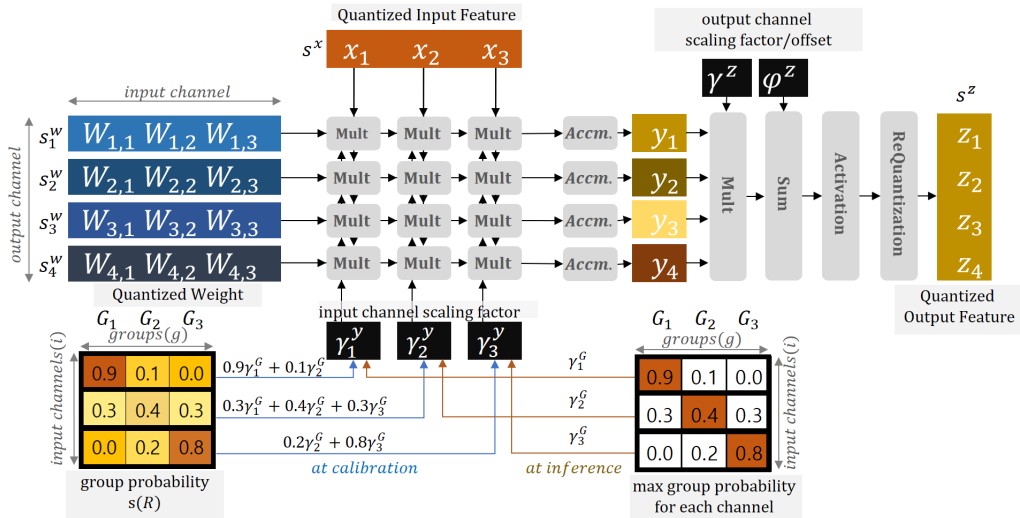

Figure 3: Overview of the calculation process for the proposed IOSO. Different colors indicate that each value possesses a distinct quantization scale.

## 3 PROPOSED METHOD

The distribution of the multiplied quantized weights and inputs can diverge from the actual values due to accumulated rounding and clipping errors from the clamp function, as well as quantization errors from the previous layers (Nagel et al., 2019; Finkelstein et al., 2019; Sun et al., 2019; Hubara et al., 2020). The approaches presented in (Nagel et al., 2020; Yuan et al., 2021; Li et al., 2021; Wei et al., 2022) sought to minimize the sum of squared differences between original output features and quantized features by solely adjusting weight rounding. Nevertheless, as quantization modifies both the weights and input features, solely relying on rounding in low-bit quantization poses limitations in achieving close approximations to the full-precision values.

Fig. 2 shows an example of 1-bit weight quantization with $\bar{w} \in \{\pm 1\}$ combined with 2-bit feature quantization with $\bar{x} \in \{0, 1, 2, 3\}$. The four blue bars represent the product of $\hat{w}$ and $\hat{x}$. In this example, $w$ is negative, and $\hat{w}$ is quantized as $-1$. During calibration, we train the offset parameter $\varphi$, which shifts the distribution of $\hat{x}\hat{w}$ by an offset. Simultaneously, the scale parameter $\gamma$ is adjusted to modify the spacing between consecutive quantized values. Both $\gamma$ and $\varphi$ are trained channel-wise to find the quantization distribution that best represents the actual value's distribution. The process of weight rounding optimization sets specific values for the weights to be used during inference. In other words, $\varphi$ and $\gamma$ are used to find the optimal distribution that quantized $\hat{w}\hat{x}$ represents, channel-wisely, and the weight rounding optimization ultimately determines which weights are used, on a weight-by-weight basis. In our implementation, $\varphi$, $\gamma$, and rounding are trained simultaneously to ensure an accurate reconstruction of the full precision value.

Our proposed method, *Input- and Output-channelwise Scaling and Offset* (IOSO), illustrated in Fig. 3 reduces this distributional mismatch, thereby mitigating the quantization error without much computational overhead (less than 1.5%) as will be shown below. Comparison with related works are included in Appendix B.

### 3.1 INPUT CHANNEL-WISE SCALE GROUPING (ISG)

On top of the output channel-wise quantization, we use different scales along with $c$ input channels. As described in Sec. 2.1, employing output channel-wise quantization allows the representation of the sum of quantized values as an affine form of full precision values $\alpha_k \sum_i^c \bar{w}_{ki}\bar{x}_i + \beta_k$. By introducing per-input channel scaling factors, denoted as $\gamma^y \in \mathbb{R}^c$, it becomes $\alpha_k \sum_i^c (\gamma_i^y \bar{w}_{ki}\bar{x}_i) + \beta_k$.

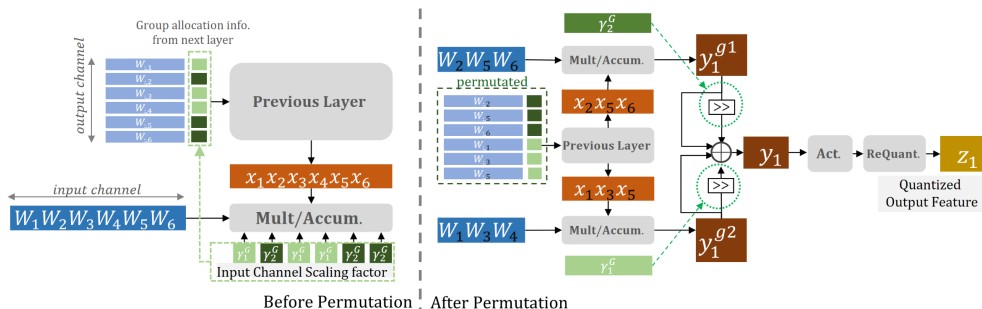

Figure 4: Comparison of computation processes before and after the permutation of weights from the previous layer.

However, incorporating a distinct scale value for each input channel inevitably introduces floating point operations due to different resolutions in different input channels. This leads to a significant increase in computational complexity. To address this issue, we use ISG, which separates input channels into multiple groups of different scales and searches for the optimal grouping on an input channel-by-channel basis. Each input channel belongs to one of the scale groups, $\{G_1, \cdots, G_g\}$ with scale values $\{\gamma_1^G, \cdots, \gamma_g^G\}$. The channels within each group are configured to carry out integer multiplication and accumulation operations together.

### 3.2 OUTPUT CHANNEL-WISE SCALE AND OFFSET (OSO)

While AdaQuant (Hubara et al., 2020) learns only the output channel-wise offset, our model simultaneously learns the scale and offset for better reconstruction at low bits. The data is affinely transformed before re-quantization using the trainable scale $\gamma^z \in \mathbb{R}^m$ and offset $\varphi^z \in \mathbb{R}^m$ resulting in $z_k = \gamma_k^z \alpha_k \sum_i^c (\gamma_i^y \bar{w}_{ki} \bar{x}_i) + \gamma_k^z \beta_k + \varphi_k^z$. OSO parameters, $\gamma^z$ and $\varphi^z$ adjust the distribution of the activation outputs to better align with the real values before the activation function and re-quantization. By making the scaling factor and offset values trainable, the model can learn the optimal reconstruction parameters during the calibration process, thereby improving the quantization accuracy and the quantized model's overall performance.

### 3.3 COMPUTATIONAL COMPLEXITY AT INFERENCE

**Precomputation** To perform accumulation within each group, computing each layer becomes

$$z_k = \gamma_k^z \alpha_k \sum_p^g \gamma_p^G \sum_{i \in G_p} \bar{w}_{ki} \bar{x}_i + \gamma_k^z \beta_k + \varphi_k^z = \alpha_k' \sum_p^g \gamma_p^G \sum_{i \in G_p} \bar{w}_{ki} \bar{x}_i + \beta_k'. \tag{6}$$

In Eq. (6), the last two terms in the middle are merged into one variable $\beta_k'$. The real-typed variables $\gamma^z \in \mathbb{R}^m$ and $\varphi^z \in \mathbb{R}^m$ are known before inference time, thus $\alpha_k'(= \gamma_k^z \alpha_k)$ and $\beta_k'(= \gamma_k^z \beta_k + \varphi_k^z)$ can be pre-computed. To reduce the computational cost associated with high-cost floating-point operations, we set each $\gamma^G \in \mathbb{R}^g$ value to be computable by integer shift/sum operations, using values such as $1.0 \pm 2^{-n}$, where $n \in \{0, 1, 2, ...\}$.

**Channel Permutation** When performing accumulation across input channels, the hardware may not perform optimally if the values are mixed in an unstructured arrangement, and additional memory is needed to store information about the selected groups.

As depicted in Fig. 4, by pre-adjusting the order of the output channels of the preceding layer, we can alter the order of the input channels for the subsequent layer. If we handle this by performing operations in independent layers per group of input channels and adding independent accumulation values at the accumulation point, we can achieve computational speeds nearly identical to the original speeds, provided that there are not many groups and no excessively small groups.

**Computational Cost per Approach** Table 1 illustrates the computational cost associated with each approach for generating a single output feature. In the absence of quantization, the convolution layer

with $k_h \times k_w$ kernel necessitates $k_h \times k_w \times c$ multiplication operations and $k_h \times k_w \times c - 1$ addition operations, executed via the MAC process. The conventional *output channel-wise quantization* (CW-Quant.) requires integer operation for MAC, coupled with an additional floating MAC operation due to re-quantization. The per-output channel adjusting scale and offset method proposed in Sec. 3.2 incurs no additional computational cost with precomputation.

When standard *input channel-wise quantization* (ICWQ) is incorporated, operations are executed in the floating-point domain, which invariably leads to an increase in computation. ICWQ seems like a good compromise because it only requires the number of input channels but is excessively expensive as 32-bit floating multiplication consumes x296 times the power of 4-bit integer multiplication (Horowitz, 2014). On the other hand, our ISG method, as proposed in Sec. 3.1 only necessitates a shift operation and an addition operation per group. Thus, the proposed method can present a better-performing alternative with almost no computational overhead compared to the conventional CW-Quant (See Tab. 1).

Table 1: Computational complexity: MAC operations for different approaches on ResNet-18 layer1.conv1 layer, which have $[3 \times 3 \times 64]$ weights per one quantized output feature. ($g$ is the number of groups for our ISG method.)

| Approach | Floating Point | | Integer | | Other |
|---|---|---|---|---|---|
| | Mul. | Add. | Mul. | Add. | Shift |
| FP32 model | 3x3x64 | 3x3x64-1 | - | - | - |
| CW-Quant. | 1 | 1 | 3x3x64 | 3x3x64-1 | - |
| +OSO | 1 | 1 | 3x3x64 | 3x3x64-1 | - |
| +ICWQ | 64 | 64-1 | 3x3x64 | - | - |
| +ISG | 1 | 1 | 3x3x64 | 3x3x64-1+$g$ | $g$ |

### 3.4 OPTIMIZATION PROCESS

**Weight Quantization** For the purpose of weight quantization, a rounding policy introduced in Adaround (Nagel et al., 2020) is adopted,

$$\bar{W}_k = Q(W_k) = \texttt{clamp}(\lfloor W_k/s \rceil + h(V_k); -2^{n-1}, 2^{n-1} - 1), \tag{7}$$

$$\text{where }, h(V_k) = \texttt{clamp}(\sigma(V_k)(\zeta - \tau) + \tau; 0, 1). \tag{8}$$

Here, $V \in \mathbb{R}^{c \times m}$ is a learnable parameter utilized for rounding, while the constants $\zeta$ and $\tau$ are employed for stretching the sigmoid function $\sigma(\cdot)$ and are conventionally set to 1.1 and $-0.1$ respectively. The function $h(V)$, introduced in (Louizos et al., 2017), guides values to converge towards either 0 or 1. It employs a clipping mechanism to prevent differentiation at extreme values, thus guaranteeing convergence to either 0 or 1. During the calibration process, this function plays a critical role in minimizing the reconstruction error between rounding down and up. Instead of the conventional on/off rounding in standard quantization, the floor function is employed with the on/off rounding being learned via the rectified sigmoid function, $h(V)$.

**Input channel-wise Scale Grouping** During calibration, the scale $\gamma_i^y$ of the ISG for the input channel $i$ is calculated and used as follows,

$$\gamma_i^y = \sum_p^g \left( \gamma_p^G s(R_{ip}) / \sum_j^g s(R_{ij}) \right), \tag{9}$$

$$s(R_{ip}) = \texttt{clamp}\left( \frac{\sigma(R_{ip})}{\sum_j^g \sigma(R_{ij})}(\zeta - \tau) + \tau; 0, 1 \right), \tag{10}$$

where the learnable parameters $R \in \mathbb{R}^{c \times g}$ are used to find the probability that the input channel belongs to a certain scale group $G_p^{(l)}$. The rectified softmax function $s(R_{ip})$ is utilized to express the probability of inclusion in each group. Similar to $h(V)$, if the probabilities are not fixed during calibration, as in Fig. 3 and Eq. (9), the $\gamma_G^p$ values of the different groups are combined in proportion to their probabilities. However, if the value of $s(R_{ip})$ reaches 1, the $\gamma_i^y$ is fixed to a specific $\gamma_p^G$ value.

**Regularization** The values of $h(V)$ used in weight quantization and the value of $s(R_{ip})$, which represents the probability of inclusion in each group in input channel grouping, should converge to 0 or 1 during calibration with regularization term presented in (Nagel et al., 2020) as follows:

$$f_{reg}(V, R; \lambda_r, \lambda_g) = \lambda_r \sum_i^m \sum_j^c (1 - |2h(V_{ij}) - 1|^\beta) + \lambda_g \sum_i^c \sum_p^g (1 - |2s(R_{ip}) - 1|^\beta), \tag{11}$$

Table 2: Evaluation on weight only quantization (top-1 accuracy(%)) on the ImageNet validation set. * represents the numbers are from BRECQ.

| Methods | Bits(W/A) | ResNet-18 | ResNet-50 | MobileNetV2 | RegNet-600MF | RegNet-3.2GF | MnasNet-2.0 |
|---|---|---|---|---|---|---|---|
| FullPrec. | 32/32 | 71.08 | 77.00 | 72.49 | 73.71 | 78.36 | 76.68 |
| AdaRound* | 4/32 | 68.71 | 75.23 | 69.78 | 71.97 | 77.12 | 74.87 |
| AdaQuant* | 4/32 | 68.82 | 75.22 | 44.78 | - | - | - |
| BRECQ* | 4/32 | 70.70 | 76.29 | 71.66 | 73.02 | 78.04 | 76.00 |
| IOSO(Ours) | 4/32 | $\mathbf{70.72}_{\pm 0.08}$ | $\mathbf{76.37}_{\pm 0.09}$ | $\mathbf{72.08}_{\pm 0.05}$ | $\mathbf{73.16}_{\pm 0.06}$ | $\mathbf{78.20}_{\pm 0.05}$ | $\mathbf{76.13}_{\pm 0.04}$ |
| AdaRound* | 3/32 | 68.07 | 73.42 | 64.33 | 67.71 | 72.31 | 69.33 |
| AdaQuant* | 3/32 | 58.12 | 67.61 | 12.56 | - | - | - |
| BRECQ* | 3/32 | 69.81 | 75.61 | 69.50 | 71.48 | 77.22 | 74.58 |
| IOSO(Ours) | 3/32 | $\mathbf{70.02}_{\pm 0.11}$ | $\mathbf{75.77}_{\pm 0.06}$ | $\mathbf{70.43}_{\pm 0.08}$ | $\mathbf{72.02}_{\pm 0.13}$ | $\mathbf{77.60}_{\pm 0.07}$ | $\mathbf{74.96}_{\pm 0.09}$ |
| AdaRound* | 2/32 | 55.96 | 47.95 | 32.54 | 25.66 | 24.70 | 30.60 |
| AdaQuant* | 2/32 | 0.30 | 0.49 | 0.11 | - | - | - |
| BRECQ* | 2/32 | 66.30 | 72.40 | 59.67 | 65.83 | 73.88 | 67.13 |
| IOSO(Ours) | 2/32 | $\mathbf{67.22}_{\pm 0.11}$ | $\mathbf{72.96}_{\pm 0.13}$ | $\mathbf{62.16}_{\pm 0.24}$ | $\mathbf{67.01}_{\pm 0.29}$ | $\mathbf{74.97}_{\pm 0.07}$ | $\mathbf{68.18}_{\pm 0.17}$ |

where $\lambda_r$ and $\lambda_g$ are regularization hyperparameters. We anneal $\beta$ following (Nagel et al., 2020): Initially, we assign a high beta value to allow only values at the extremes of 0 or 1 to converge. Towards the end of the calibration, we implement normalization by assigning lower beta values to allow values near 0.5 to converge as well.

**Loss Function for optimization** To identify the optimal parameters $V$, $R$, $\gamma^z$s and $\varphi^z$ using a small calibration set, we utilize the block reconstruction loss as described in Sec. 2.2. Our optimization proceeds at the block level, and parameters $V$ and $R$ are subjected to regularization. The final loss function can be represented as

$$\operatorname*{arg\,min}_{V,R,\gamma^z,\varphi^z} \left\| Z - \hat{Z} \right\|_F^2 + f_{reg}(V, R; \lambda_r, \lambda_g), \tag{12}$$

where $Z$ represents the real-valued output feature while $\hat{Z}$ denotes the approximated reconstructed output feature from Eq. (6), and $\|\cdot\|_F^2$ denotes the Frobenius norm. Note that $\gamma^G \in \mathbb{R}^g$ is a hyperparameter which determines $\gamma^y$ value.

## 4 EXPERIMENTS

We evaluate the proposed method on various vision models to assess its performance. Our code is based on an open-source BRECQ (Li et al., 2021) for our implementation. To minimize the impact of hyperparameters, we set $\lambda_r = \lambda_g$. Since too much fragmentation could lead to inefficient processing in hardware, we designated the number of groups to be 3 and the group scale values $\gamma^G$ were set to (1.0, 1.0-$2^{-4}$, 1.0+$2^{-4}$). Consistent with BRECQ, we used a randomly selected calibration set of 1,024 samples from the ImageNet (Russakovsky et al., 2015) training set and the first and last layers employed 8-bit precision like (Li et al., 2021; Jeon et al., 2022; Wei et al., 2022; Zheng et al., 2022), and the weight tuning method was configured identically to that of AdaRound (Nagel et al., 2020). The calibration process involves 35,000 iterations, taking around 40 minutes to calibrate a ResNet-18 model on a single RTX 3090 GPU.

### 4.1 MAIN RESULTS

Table 2 and Table 3 show the results of weight-only and weight-feature quantization experiments conducted on various CNN-based architectures. We conducted quantization on a variety of network architectures, including ResNet-18,50 (He et al., 2016), MobileNetV2 (Sandler et al., 2018), MNasNet_v2 (Tan et al., 2019), and RegNet (Radosavovic et al., 2020). We report the mean and the standard deviation on five trials for ours.

For 2-bit weight quantization, our proposed method improves by approximately 0.5-1.5%p compared to the baseline methods AdaRound and BRECQ. As the quantization bit increases to 3-bit and 4-bit, the difference in performance gradually decreases. This can be attributed to the fact that as

Table 3: Evaluation of top-1 accuracy (%) in weight and feature quantization; * denotes the number are from BRECQ (Li et al., 2021)

| Methods | Bits(W/A) | ResNet-18 | ResNet-50 | MobileNetV2 | RegNet-600MF | RegNet-3.2GF | MnasNet-2.0 |
|---|---|---|---|---|---|---|---|
| FullPrec. | 32/32 | 71.08 | 77.00 | 72.49 | 73.71 | 78.36 | 76.68 |
| ZeroQ* | 4/4 | 21.71 | 2.94 | 26.24 | 28.54 | 12.24 | 3.89 |
| LAPQ* | 4/4 | 60.3 | 70.0 | 49.7 | 57.71 | 55.89 | 65.32 |
| AdaQuant* | 4/4 | 67.5 | 73.7 | 34.95 | - | - | - |
| BRECQ* | 4/4 | 69.60 | 75.05 | 66.57 | 68.33 | 74.21 | 73.56 |
| IOSO(Ours) | 4/4 | **69.64**$_{\pm0.06}$ | **75.12**$_{\pm0.07}$ | **67.88**$_{\pm0.15}$ | **70.54**$_{\pm0.08}$ | **76.39**$_{\pm0.08}$ | **73.63**$_{\pm0.05}$ |
| ZeroQ* | 2/4 | 0.08 | 0.08 | 0.10 | 0.10 | 0.05 | 0.12 |
| LAPQ* | 2/4 | 0.18 | 0.14 | 0.13 | 0.17 | 0.12 | 0.18 |
| AdaQuant* | 2/4 | 0.21 | 0.12 | 0.10 | - | - | - |
| BRECQ* | 2/4 | 64.80 | 70.29 | 53.34 | 59.31 | 67.15 | 63.01 |
| IOSO(Ours) | 2/4 | **66.00**$_{\pm0.05}$ | **71.09**$_{\pm0.19}$ | **56.10**$_{\pm0.12}$ | **63.60**$_{\pm0.25}$ | **72.08**$_{\pm0.08}$ | **63.42**$_{\pm0.13}$ |

the quantization bit becomes sufficiently large, the accumulated bits can be adequately represented, reducing the importance of adjusting the scaling and offset. Additionally, our method shows better results than BRECQ in experiments with 2-bit weight and 2 or 4-bit feature quantization. We also achieved up to 5% absolute improvement in feature and weight quantization than BRECQ.

We also explored the combination of IOSO with recent rounding-only methods, as detailed in Appendix D.1. Our method, applied to three different techniques, particularly in low bits, demonstrated performance improvements of up to 9.3%p. In Appendices D.2 and D.3, which were carried out on detection tasks and the VIT-B model, we achieved a maximum increase of 0.011 in mAP compared to other low-bit quantization methods, and we also observed a performance improvement of 0.64%p in the VIT-B model with 6-bit weight activation.

## 4.2 ABLATION STUDY

We investigate the effect of each proposed component using ResNet-18 on the W2A4 settings.

Table 4: Ablation study of OSO and ISG on the ResNet-18 model using ImageNet calibration data. Top-1 accuracy comparison on ImageNet test dataset.

**Effect of OSO** We compare the performance with and without each parameter to examine the effect of scale factor $\gamma^z$ and offset $\varphi^z$. As shown in Table 4, scale $\gamma^z$ helped to improve performance more than offset $\varphi^z$. In all cases, accuracy was highest when both variables were learned.

| Method | V | $\gamma^z$ | $\varphi^z$ | $\gamma^y$ | R | W Quant. | W&A Quant. |
|---|---|---|---|---|---|---|---|
| Round-Only | O | | | | | 66.41(+0.00) | 64.80(+0.00) |
| Out. Offset Only | O | | O | | | 66.65(+0.24) | 65.59(+0.79) |
| Out. Scale Only | O | O | | | | 67.02(+0.59) | 65.79(+0.99) |
| Out. Off.&Scale | O | O | O | | | 67.04(+0.63) | 65.87(+1.07) |
| Inp. Scale | O | | | O | | 67.18(+0.77) | 66.04(+1.24) |
| +Out. Off.&Scale | O | O | O | O | | **67.21**(+0.80) | **66.05**(+1.25) |
| Inp. Scale Grouped | O | | | | O | 67.02(+0.61) | 65.88(+1.08) |
| +Out. Off.&Scale | O | O | O | | O | **67.19**(+0.78) | **65.97**(+1.17) |

**Effect of Grouping** To compare the effect of grouping, we looked at the results with and without grouping when applying ICWS. Without grouping, we saw the highest accuracy because the model could find the optimal scale value $\gamma^y$ for each input channel. However, we can see that even with grouping, the performance drop is less than 0.1%p, which shows that the proposed ISG does not significantly sacrifice performance.

## 4.3 INPUT CHANNEL GROUP

**Input Channel Group Granularity** While adjusting OSO is a cost-free method to add, ISG has the disadvantage that hardware performance can be degraded if too many groups become fragmented. Hardware architectures designed for MAC operations, such as the systolic array, perform a specific amount of MAC operations on input and output channels simultaneously. This feature is crucial for computationally intensive tasks. Server hardware like Tensor Processing Units (TPUs (Jouppi et al., 2017)) can perform 256x256 operations simultaneously, while deep learning accelerators for edge devices can handle at least 16x16 operations (Genc et al., 2019; Wang et al., 2016) at a time. This indicates that if the input channel grouping fragments the channels, the hardware cannot efficiently utilize MAC in the direction of the input channels.

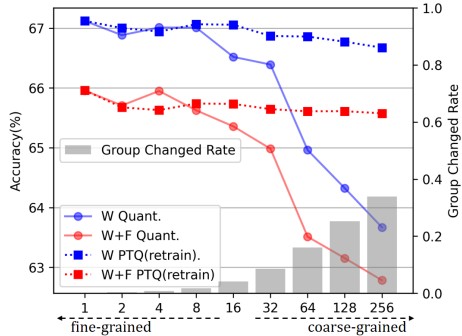 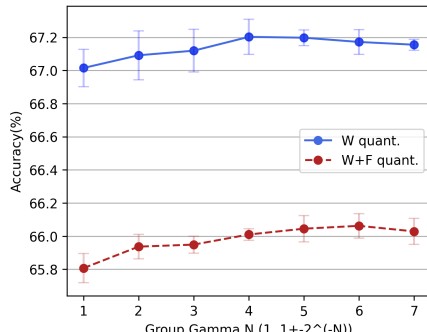

Figure 5: Performance change(top-1 accuracy) on ResNet-18 model using ImageNet for granularity variation of the input channel group.

Figure 6: Accuracy comparison on ResNet-18 model using ImageNet as the $\gamma^G$ value changes.

Our initial approach was to evaluate the accuracy by varying the granularity of the input channel groups, with grouped channels prioritized. Higher granularity signifies that the number of channels per group is not limited. That is, each channel can be assigned to its optimal input scale group $G_g$. On the other hand, coarse granularity means that the number of channels per group is more balanced, making it more compatible with certain hardware. As depicted in Fig. 5, we observed that decreasing the granularity led to a rise in the divergence from the optimal group (learned when granularity is the highest). However, by leaving the granularity-constrained groups intact and fine-tuning the remaining variables, we outperformed the traditional rounding method even at a granularity of 256.

**Input Channel Scale Factor** In Sec 3, we showed that multiplying the $\hat{w}\hat{x}$ by a $\gamma^y$ value can bring it closer to the real value distribution. To see how performance varies with $\gamma^y$, we varied $\gamma^y$ group from $(1, 1 \pm 2^{-1})$ to very small scale values such as $(1, 1 \pm 2^{-7})$ and compared the accuracy. As shown in Fig. 6, the weights quantization result was best when changing the value of $\hat{w}\hat{x}$ by about $6\%(2^{-4})$, and weight&feature quantization cases improve accuracy when changing $\hat{w}\hat{x}$ by about $1.5\%(2^{-6})$.

## 4.4 COMPUTATION COSTS FOR MODELS

To calculate the cost of each model, we counted the number of integer operations in the fully-connected and convolutional layers of each model. We then evaluated the increase in computation due to ISG compared to the baseline. We also counted the shift operation as a single integer operation. For depth-wise convolution (Howard et al., 2017), ISG is unnecessary because each input channel corresponds to an individual output channel, and each channel can use $\gamma^z$ individually. Except for MobileNetV2, the computation cost increased by less than 1.3%. The cost saving of ISG is greater when the convolution filter size is large. MobileNetV2 mostly uses 1x1 convolution, it increased by about 4% compared to the other models. However, even this

Table 5: Increased giga integer operations by ISG for each model

| Model | Baseline | ISG | Total |
|---|---|---|---|
| ResNet-18 | 3.005 | 0.006 | 3.011(0.20%) |
| ResNet-50 | 5.576 | 0.013 | 5.589(0.24%) |
| MobileNetV2 | 0.338 | 0.014 | 0.345(4.03%) |
| RegNet-600MF | 0.777 | 0.010 | 0.782(1.28%) |
| RegNet-3.2GF | 4.442 | 0.030 | 4.472(0.68%) |
| MnasNet-2.0 | 2.217 | 0.025 | 2.242(1.11%) |

number can be made much smaller for hardware that supports shift operations since shift operations can be made with simple wire connections compared to addition operations.

## 5 CONCLUSION

In this paper, we introduced IOSO, a novel post-training quantization. IOSO advances feature reconstruction by finely adjusting scales and offsets in both input and output channel-wisely. Our research shows that IOSO can be computed using simple integer operations, requiring less than a 1.5% increase in computational resources and, importantly, without additional memory. When applied to a variety of CNN-based architectures, IOSO significantly improved the accuracy of quantizing 2/4-bit weights/features by 1-5%p over rounding-only methods.

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

## A PRELIMINARIES

### A.1 BATCH-NORMALIZATION FUSING

When the weights are symmetrically quantized, and the features are asymmetrically quantized, there can be $y_k$ features per output channel of the layer,

$$y_k = \sum_i^c \hat{w}_{ki} \hat{x}_i = \sum_i^c (s_k^w \bar{w}_{ki}) \cdot (s^x (\bar{x}_i - z^x)) = s_k^w s^x \sum_i^c \bar{w}_{ki} \bar{x}_i - z^x s_k^w s^x \sum_i^c \bar{w}_{ki}, \quad (13)$$

when $y_k$ passes the batch norms and re-quantization is performed again, it becomes $z_k$,

$$\bar{z}_k = Q(\texttt{BatchNorm}(y_k)) = \texttt{clamp}\left(\left\lfloor\left(\left(\gamma_k \frac{y_k - \mu_k}{\sqrt{\sigma_k^2 + \varepsilon}} + \delta_k - z^z\right)\bigg/ s^z\right\rceil ; 0, 2^n - 1\right), \quad (14)$$

$$= \texttt{clamp}\left(\left\lfloor\alpha_k \sum_i^c \bar{w}_{ki} \bar{x}_i + \beta_k\right\rceil ; 0, 2^n - 1\right), \quad (15)$$

$$\text{where,} \quad \alpha_k = \frac{\gamma_k s_k^w s^x}{s^z \sqrt{\sigma_k^2 + \varepsilon}} \quad \text{and} \quad \beta_k = -\frac{\gamma_k (z^x s_k^w s^x \sum_i^c \bar{w}_{ki} + \mu_k)}{s^z \sqrt{\sigma_k^2 + \varepsilon}} + \frac{\delta_k - z^z}{s^z}. \quad (16)$$

Here, $\mu_k$ and $\sigma_k^2$ are the empirical mean and variance of a batch, and $\gamma_k$ and $\delta_k$ are the learnable parameters for batch normalization. By fusing with batch normalization, $\bar{z}_k$ can be calculated using *multiply-accumulate* (MAC) operations on quantized $x_i$, which are not available before inference. However, channel-wise parameters $\alpha_k$ and $\beta_k$ can be prepared before inference time. This process significantly optimizes the computational load during inference, improving efficiency. When combined with Eq. (13), batch normalization operates as a per-output channel affine function, allowing $y_k$ to be expressed as an affine form of the accumulated quantized values $\alpha_k \sum_i^c \bar{w}_{ki} \bar{x}_i + \beta_k$.

## B RELATED WORK

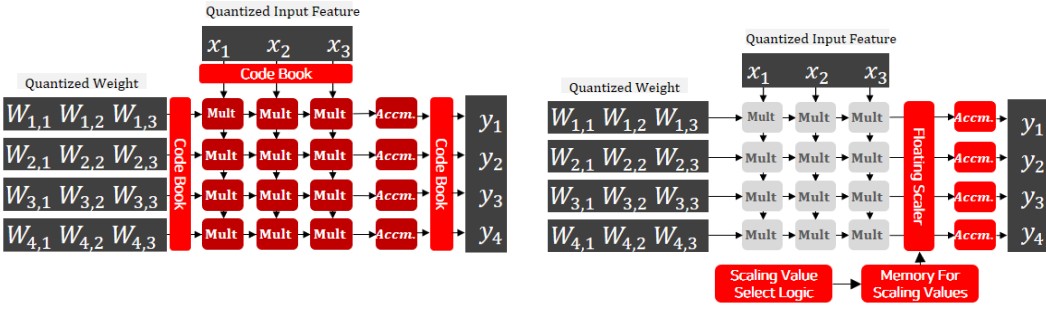

(a) Systolic array using codebook.          (b) Systolic array using input scaling.

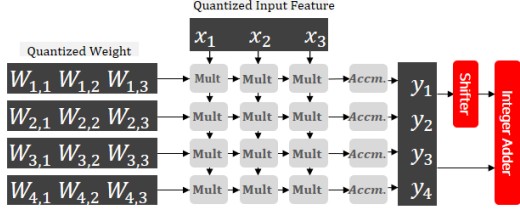

(c) Systolic array of our method.

Figure 7: Comparison of a systolic array of each quantization method.

To the best of our knowledge, we have developed an approach that enhances performance with a limited dataset (PTQ) through channel grouping and solely using shift operations without modifying the internals of the systolic array. We have compared the methods related to our previously researched approach. Fig. 7 depicts the changes in the systolic array for each method. The gray area represents the original integer systolic array, while the red area indicates the portions that change when each method is applied. Our method does not alter the internal structure of the systolic array.

**Reconstruction Error Minimize** Previous studies, which aimed to minimize reconstruction errors, employed techniques such as rounding up or down weights methods (Nagel et al., 2020; Li et al., 2021; Wei et al., 2022; Zheng et al., 2022) to find optimal values. AdaQuant (Hubara et al., 2020) took it further by using the optimal batch normalization bias. However, in extremely low-bit quantization scenarios, simply relying on weight rounding and bias proved inadequate for reconstruction. Mr.BiQ (Jeon et al., 2022) employed non-uniform quantization to improve reconstruction, but this non-linear approach posed challenges for efficient utilization on standard hardware. We further scale and offset the input and output channels to reduce the reconstruction error to minimize the distributional gap.

**Quantization Grouping** Some studies (Stock et al., 2019; Nascimento et al., 2019; Darvish Rouhani et al., 2020) attempted to optimize the quantization process by extending the scaling group beyond output channel-wise to the input channels of the model. However, such an approach necessitates using additional memory or floating-point operation units, significantly increasing power consumption compared to integer units. Our approach maintains the most computationally intensive parts of the systolic array in integers without requiring additional memory usage. Furthermore, instead of computationally complex floating-point operations, it only necessitates integer shift operations outside the systolic array.

**Weight/Feature Scaling** FlexRound (Lee et al., 2023) placed a learnable parameter that can directly manipulate the pre-trained weight to modify the weight. Both Data-Free Quantization Through Weight Equalization and Bias Correction (Nagel et al., 2019) and Smooth Quantization (Xiao et al., 2023) reduced the size of the Activation and increased the corresponding weight scale to perform quantization. These methods transform existing input activations and weights into a shape suitable for quantization. We aimed to keep the variation of the computed product as close to FP32 as possible without altering the activation and weight as Fig. 7c.

**Weight Step Size Learning** Research like LSQ (Esser et al., 2019) brought performance improvement by learning the quantization step size and did not approach from the perspective of the product of weight and feature. Moreover, this approach entails using all datasets and also entails a lengthy quantization process QAT.

**Output Channel-wise biasing** AdaQuant (Hubara et al., 2020) applied output channel-wise biasing. As can be seen in Table 4. In the Ablation Study, our output channel-wise offset also plays an additional role in mitigating the biasing that can occur through Input Channel Grouping.

**Input Channel-wise scaling** SYQ (Faraone et al., 2018) proposed a method to learn the pixel/row-wise scaling value of the convolution weight, but as Fig. 7b it requires using an FP32 accumulator instead of an integer accumulator due to the use of real-value scaling values. Compared to FP32 multiplication, but our IOSO, integer shift operations use 24 times less power on ASIC (45nm) and 196 times less on FPGA (ZYNQ-7 ZC706) (Horowitz, 2014).

## C  ALGORITHM

Algorithm 1, shows the overall parameter learning process of IOSO. First, weight quantization is performed, which optimizes the full-precision model block by block. In weight quantization, the parameters that minimize block reconstruction are learned through $V$, which determines the rounding policy, $\gamma^z$, $\varphi^z$, which determines the output channel-wise scale, offset, and $R$, which determines the input channel-wise scale group. After all the blocks in the model are weight quantized, we proceed to feature quantization. In feature quantization, the scale value of the feature $s^x$ is determined using a *straight-through estimator* (STE) because it is not related to the rounding policy parameter of the weight. We do not learn $R$, which determines the input channel-wise group, because it has already converged in the weight quantization step. The $\gamma^z$ and $\varphi^z$ for fitting the offset and scale are still trained to complete the feature quantization.

---

**Algorithm 1** IOSO Post training quantization

---

1: **Inputs :** *full-precision Model*, *Calibration Set*
2: **procedure** IOSO_PTQ
3:     **Step 1:** Weight Quantization
4:     *Model ← full-precision Model*
5:     **for** *block* **in** *Model* **do**
6:         Minimize *block* reconstruction loss with $V$, $\gamma^z$, $\varphi^z$, $R$
7:         *block ←* reconstructed *block*
8:     **end for**
9:     **Step 2:** Feature Quantization
10:     **for** *block* **in** *Model* **do**
11:         Minimize *block* reconstruction loss with $s^x$, $\gamma^z$, $\varphi^z$
12:         *block ←* reconstructed *block*
13:     **end for**
14:     **return** $Model$
15: **end procedure**

---

Table 6: A comparison of quantization performance when applying IOSO to BRECQ (Li et al., 2021), QDROP (Wei et al., 2022), and NWQ (Zheng et al., 2022).

| Quantized Bits | | method | resnet18 | resnet50 | mobilenetv2 | regnetx600m | regnetx3200m | mnasnet |
|---|---|---|---|---|---|---|---|---|
| weight | activation | | | | | | | |
| 4 | 4 | BRECQ | **68.934** | **74.846** | 67.432 | **70.438** | 76.404 | 72.328 |
| | | **BRECQ+IOSO** | 68.826 | 74.806 | **67.864** | 70.312 | **76.412** | **72.556** |
| | | QDrop | **69.138** | 74.990 | 67.946 | 70.870 | 76.456 | 73.042 |
| | | **QDrop+IOSO** | 69.104 | **75.116** | **68.470** | **71.004** | **76.614** | **73.206** |
| | | NWQ | **69.026** | 74.810 | 67.668 | **70.574** | 76.398 | **72.632** |
| | | **NWQ +IOSO** | 68.960 | **74.954** | **68.000** | 70.438 | **76.584** | 72.596 |
| 2 | 4 | BRECQ | 63.922 | 69.582 | 52.368 | 61.532 | 71.050 | 60.424 |
| | | **BRECQ+IOSO** | **64.572** | **70.452** | **54.414** | **62.804** | **71.846** | **61.428** |
| | | QDrop | 64.462 | 69.710 | 53.608 | 62.806 | 71.870 | **61.876** |
| | | **QDrop+IOSO** | **65.198** | **70.790** | **55.446** | **63.852** | **72.474** | 61.870 |
| | | NWQ | 64.508 | 69.768 | 53.474 | 62.328 | 71.604 | 61.260 |
| | | **NWQ +IOSO** | **64.828** | **70.374** | **54.830** | **63.030** | **72.200** | **62.256** |
| | 2 | BRECQ | 47.616 | **48.640** | 5.118 | **27.878** | **41.784** | 10.310 |
| | | **BRECQ+IOSO** | **47.664** | 47.994 | **8.562** | 27.022 | 37.854 | **19.668** |
| | | QDrop | 51.960 | 55.442 | 10.108 | 39.336 | 54.586 | 21.970 |
| | | **QDrop+IOSO** | **52.976** | **55.874** | **15.344** | **41.264** | **55.862** | **26.454** |
| | | NWQ | 49.742 | 52.538 | 9.066 | 33.746 | 50.810 | 20.854 |
| | | **NWQ +IOSO** | **50.872** | **52.770** | **10.672** | **35.494** | **51.196** | **27.162** |
| 3 | 3 | BRECQ | **64.980** | 70.324 | 51.304 | 62.626 | **70.984** | 61.594 |
| | | **BRECQ+IOSO** | 64.766 | **70.492** | **54.628** | **62.926** | 70.530 | **63.122** |
| | | QDrop | 65.616 | 71.332 | 54.836 | 64.688 | 71.764 | 64.276 |
| | | **QDrop+IOSO** | **65.774** | **71.440** | **57.010** | **65.292** | **72.328** | **64.878** |
| | | NWQ | 65.128 | 70.728 | **53.750** | 63.362 | 71.564 | 62.758 |
| | | **NWQ +IOSO** | **65.242** | **70.888** | 55.390 | **63.880** | **71.874** | **63.820** |

# D   ADDITIONAL RESULTS

## D.1   IMAGENET CLASSIFICATION WITH RECENT ROUND-BASED METHOD

Table 6 is from additional experiments conducted to see if performance improvements occur in low-bit quantization by applying IOSO to the existing latest round-based methods. We achieved performance enhancement in low-bit quantization by reducing the reconstruction error. We used the publicly available GitHub code for BRECQ (Li et al., 2021) and QDrop (Wei et al., 2022). We implemented and tested the Activation Regularization, Annealing Softmax, and Annealing Mixup as proposed in NWQ (Zheng et al., 2022).

Table 7: Performance comparison when quantizing the backbone (ResNet50) of Faster RCNN using each method

| Bits | | mAP | mAP50 | mAP75 | mAPs | mAPm | mAPl |
|------|------|------|------|------|------|------|------|
| | **BASE** | 0.403 | 0.610 | 0.44 | 0.24 | 0.441 | 0.515 |
| 4/4 | BRECQ | **0.378** | **0.583** | **0.410** | 0.219 | 0.414 | **0.489** |
| | **BRECQ+IOSO** | **0.378** | 0.581 | 0.409 | **0.222** | **0.415** | **0.489** |
| | QDROP | 0.379 | 0.584 | 0.412 | 0.220 | 0.415 | **0.489** |
| | **QDROP+IOSO** | **0.381** | **0.585** | **0.414** | **0.225** | **0.416** | **0.489** |
| 3/3 | BRECQ | 0.341 | 0.535 | 0.364 | **0.192** | 0.374 | 0.451 |
| | **BRECQ+IOSO** | **0.342** | **0.538** | **0.365** | 0.189 | **0.375** | **0.452** |
| | QDROP | 0.346 | 0.545 | 0.372 | 0.197 | 0.378 | 0.455 |
| | **QDROP+IOSO** | **0.352** | **0.552** | **0.379** | **0.203** | **0.386** | **0.458** |
| 2/4 | BRECQ | 0.357 | 0.556 | 0.386 | 0.201 | 0.391 | 0.469 |
| | **BRECQ+IOSO** | **0.365** | **0.566** | **0.392** | **0.213** | **0.400** | **0.474** |
| | QDROP | 0.360 | 0.560 | 0.387 | 0.206 | 0.396 | 0.474 |
| | **QDROP+IOSO** | **0.369** | **0.571** | **0.399** | **0.215** | **0.406** | **0.480** |

## D.2 DETECTION TASK

Table 7 is the detection task result with applying our IOSO to the backbone (resnet50) of the faster rcnn coco model. IOSO operates orthogonally to the conventional rounding scheme of PTQ research (BRECQ (Li et al., 2021), QDrop (Wei et al., 2022)). We used regularization weight 0.01 following QDrop and the result is superior or equivalent to the baselines. Especially, for extreamly low bit quantization, the performance gain is prominent.

## D.3 VIT-B MODEL(IMAGENET DATASET)

We applied IOSO to the VIT-B model using the ImageNet dataset. We set the weight and activation to 6 bits for the experiment and applied a regularization weight of 1.5. The results revealed that when both the weight and features were quantized to 6 bits, the BRECQ-only approach achieved an accuracy of 79.844%, while combining BRECQ and IOSO resulted in an accuracy of 80.482%.

When we quantized the weight to 4 bits and the activation to 6 bits, the BRECQ-only quantization achieved an accuracy of 79.826%. However, by including IOSO, we achieved an accuracy of 80.024%.

## D.4 REGULARIZATION PARAMETER LAMBDA

In our study, we established the values of $\lambda_r$ and $\lambda_g$ equal to simplify the hyperparameter tuning process. The role of $\lambda_r$ is to regularize the rounding up or down, essentially influencing the convergence rate for rounding probability. Conversely, $\lambda_g$ governs the speed at which the determination of the input channel-wise scale value reaches convergence. Since rounding and ISG mutually affect one another, the rate at which each probability converges can produce varying outcomes. It is plausible that using identical $\lambda$ values may not consistently deliver optimal results.

Fig. 8 presents the findings of an experiment conducted with differing $\lambda_r$ and $\lambda_g$ values. As the figure illustrates, maintaining identical $\lambda_r$ and $\lambda_g$ values does not necessarily guarantee optimal performance. The rounding regulating factor, $\lambda_r$, exhibits higher sensitivity, whereas most of the $\lambda_g$ values produced satisfactory results at a value of 0.01. If $\lambda_r$ is not set optimally, determining the appropriate $\lambda_g$ becomes more challenging.

## D.5 IMPACT OF OSO ON INPUT CHANNEL-WISE SCALE

Table 8 illustrates the impact of output scale $\gamma^z$ and offset $\varphi^z$ when applying Input Scale $\gamma^y$ and Input Scale Group, respectively. The baseline is established with the policy of rounding only.

When solely applying the Input Scale $\gamma^y$ without any grouping, learning the output channel offset $\varphi^z$ improved accuracy. However, when learning the output channel scale $\gamma^z$ and the Input Scale $\gamma^y$

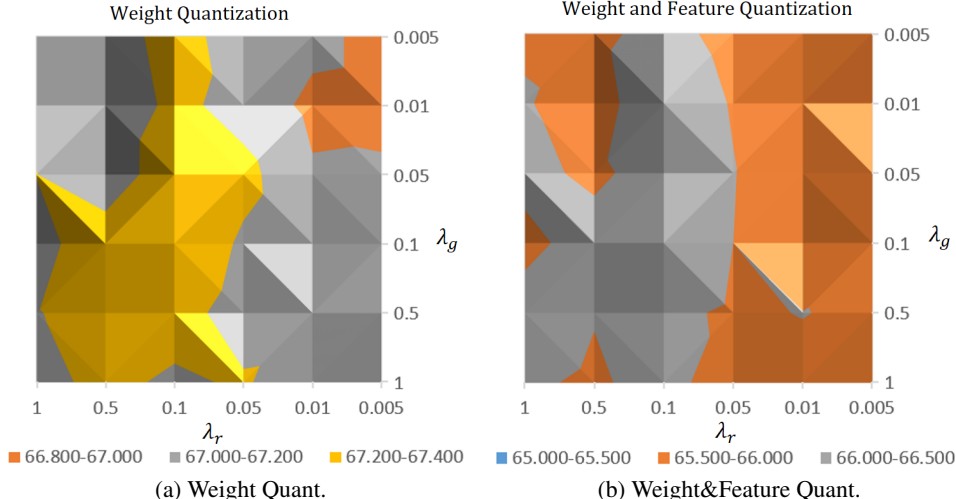

(a) Weight Quant.                    (b) Weight&Feature Quant.

Figure 8: Comparison of top-1 accuracy for ResNet-18 on the ImageNet test dataset as two $\lambda$ values vary.

Table 8: The effect of output channel-wise scale and offset on input channel-wise scale. ResNet-18 ImageNet calibration is used and is in the top 1% of accuracy.

| Method | $V$ | $\gamma^z$ | $\varphi^z$ | $\gamma^y$ | $R$ | W Quant. | W&A Quant. |
|---|---|---|---|---|---|---|---|
| Round-Only | O | | | | | 66.411(+0.000) | 64.800(+0.00) |
| Inp. Scale | O | | | O | | 67.175(+0.765) | 66.037(+1.237) |
| +Out. Offset | O | | O | O | | 67.183(+0.772) | 66.061(+1.261) |
| +Out. scale | O | O | | O | | 67.045(+0.635) | 66.001(+1.201) |
| +Out. Off.&Scale | O | O | O | O | | **67.211**(+0.800) | **66.052**(+1.252) |
| Inp. Scale Grouped | O | | | | O | 67.022(+0.611) | 65.875(+1.075) |
| +Out. Offset | O | | O | | O | 67.033(+0.623) | 65.948(+1.148) |
| +Out. Scale | O | O | | | O | 67.100(+0.690) | 65.934(+1.134) |
| +Out. Off.&Scale | O | O | O | | O | **67.192**(+0.781) | **65.973**(+1.173) |

simultaneously, it decreased accuracy. However, we achieved the best accuracy when both variables $\gamma^z$ and $\varphi^z$ were trained simultaneously.

In the case of the grouped Input Scale, there was no significant decrease in performance. However, the output channel scale $\gamma^z$ showed superior results compared to the offset $\varphi^z$. Applying an output scaling without an offset does not seem to contribute significantly. In the case of the grouped Input Scale, using only the input scale was insufficient to fit the distribution effectively. Therefore, combining the output scale and offset can lead to improved outcomes.

### D.6 INPUT CHANNEL GROUP GRANULARITY

Fig. 9 illustrates the practical group assignment procedure under changes in granularity. Initially, we trained all input channels to learn groups without group regulation. In the rectified softmax function, these groups become fixed when their probability surpasses a certain threshold. Fig. 9a exhibits the input channels that were initially fixed and the groups selected at that stage. In the ResNet-18 architecture, specifically in layer 4.1.conv0, the group initially assigned with the highest number of fixed input channels was group 2. However, as the process progressed, it turned out that group 1 ended up having the highest number of fixed input channels.

Fig. 9b demonstrates that the maximum number of input channels each group could accommodate per bundle ranged from 1, 2, 4, up to 128. As shown in Fig. 9a, the input channels that achieved convergence at a faster rate possessed greater influence and priority, leading them to be chosen first for the group. The groups determined later had less influence on reducing the loss, regardless of their

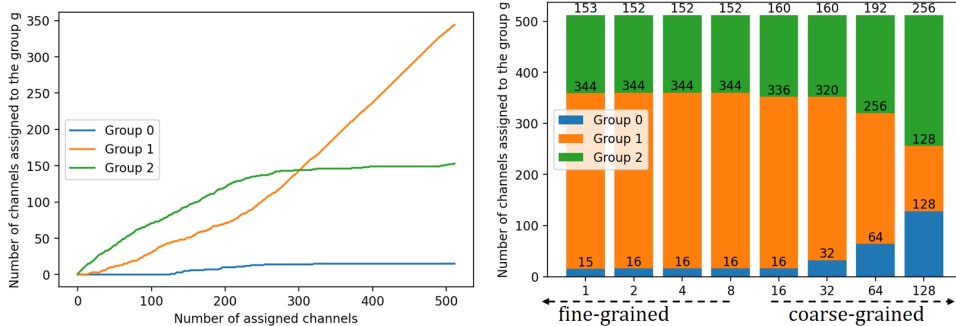

(a) Changes in granularity and the number of chan-(b) The order in which the final group was deter-
nels assigned to a group mined

Figure 9: Channel allocates for groups in the ResNet-18 model's layer4.1.conv0, to vary the granularity of input channel groups.

group. If a channel could not be assigned to its original group, it was allocated to an unoccupied group. Therefore, as depicted in Fig. 9a, despite the majority of input channels being assigned to Group 2 and the fewest to Group 0, the input channels assigned to Group 0 are first assigned 128 groups, and those assigned to Group 2 receive two sets of 128 groups. Hence, Group 1 only has a group size of 128 when the final group size is 128.

### D.7 LOCATION OF THE OSO BEFORE OR AFTER ACTIVATION FUNCTION

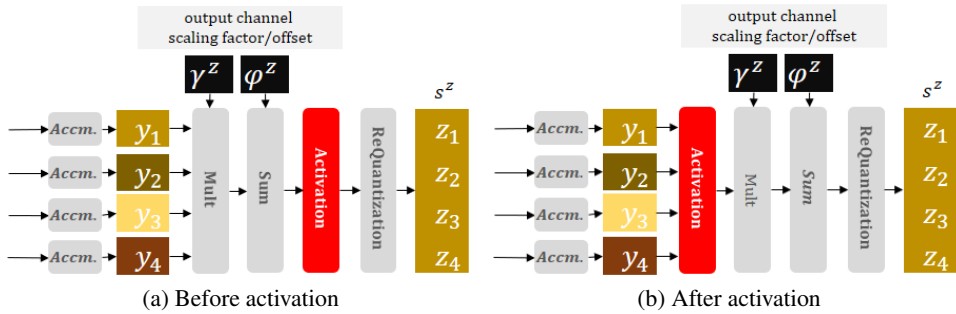

(a) Before activation            (b) After activation

Figure 10: Location of the OSO before or after activation function.

Output channel-wise scale and offset can be applied in two locations: before and after the activation function. Placing the output channel-wise scale and offset before the activation function provides the advantage of capturing a more accurate representation of the activation input, as some information may be lost after the activation. On the other hand, placing it after the activation layer allows it to be positioned closer to the re-quantization process, resulting in a better representation of the distribution of the final output features.

However, as shown in Table 9, when we moved the location of applying the output scale $\gamma^z$ and offset $\varphi^z$ after activation, we found that the performance of weight and activation quantization decreased significantly.

Table 9: Quantization results depend on the position of the OSO before and after the activation function.

|  | W Quant. | W+F Quant. |
| --- | --- | --- |
| Before Act. | 67.199 | 65.974 |
| After Act. | 67.171 | 44.718 |

We moved the output scale $\gamma^z$ and offset $\varphi^z$ to the back of the activation and tried ablation to learn the added parameter. As shown in Table 10, we can see that not learning offset $\varphi^z$ shows a similar performance to the original performance. Negative values can challenge the learning process in most activation functions because the gradient becomes zero, leading to a vanishing gradient problem. As

Table 10: The accuracy of learning each parameter when output scale $\gamma^z$ and offset $\varphi^z$ are applied after activation.

| $\gamma^z$ | $\varphi^z$ | $R$ | W Quant. | W+F Quant. |
|---|---|---|---|---|
| O | O | O | 67.140 | 45.130 |
|   | O | O | 67.070 | 50.056 |
| O |   | O | 67.152 | **65.840** |
| O | O |   | 67.210 | 51.932 |
|   | O |   | 66.774 | 49.686 |
| O |   |   | 67.078 | **66.016** |

a result, only positive values can be effectively learned. This limitation can impact the model's ability to capture and represent negative values during learning. In the feature quantization step, the scale value is learned together through the STE, but it seems that the scale $\gamma^z$ is not learned well due to the influence of the offset term and the activation function, resulting in this result.

Also, as shown in Table 9, applying the output channel scale $\gamma^z$ and offset $\varphi^z$ after activation does not significantly increase the performance of weight quantization too. So it seems that applying the output scale and offset before the activation layer can achieve better results.

### D.8    INPUT CHANNEL SCALE WITHOUT GROUP CONSTRAINTS

In our experimental setup, we implemented the same group scale across all layers to simplify the process of hyperparameter tuning. However, using the same group scale may not be optimal as each layer possesses distinctive characteristics.

To determine the optimal alpha for each layer, we enabled each channel to independently explore the optimal input channel scale value without any group restrictions. The results of this exploration are illustrated in Fig. 11, which displays the distribution of input channel scale values across different layers.

In Fig. 11, the dotted line represents our $\gamma^G$ value, which is set at $1 \pm 2^{(-4)}$. It is evident from the figure that the distribution of the input channel scale varies considerably across different layers.

In particular, for the down-sampling layer, assigning a larger group scale value seems suitable. As for the fully connected layer, altering the value of the input scale appears challenging due to the high values of the interrelated input channels.

## E    ASIC RESULTS

We conducted experiments using the structure of a batch-norm fused systolic array from an NPU developed for edge devices to compare overhead and latency in ASICs. The table 11 shows the area of the batch-norm fused systolic array when each method is implemented in ASIC. The synthesis was done using the TSMC 12nm process and 400 Mhz timing conditions. The MAC has a 16x16 structure, and IOSO, batch norm, and activation can process 8 data in 1 cycle. The sequence is MAC($\rightarrow$IOSO)$\rightarrow$ BN$\rightarrow$ACT. The performance time, or latency, was tested only on the resnet-50 layer with 28x28x128 feature 3x3x128x128 kernel layer.

As seen in the table above, IOSO can be implemented with an integer shifter adder without modifying the MAC structure, resulting in only a 2.37% increase in area. In addition, the implementation is very simple as it does not alter the existing data path. In contrast, methods partially implemented with FP16 points for increased accuracy and reduced memory are useful in GPUs that already have FP16 point kernels. However, from an ASIC perspective, these are challenging to use due to significant increases in

Table 11: ASIC synthesis area of each method and latency of convolution layer.

| Methods | integer Quant. | +IOSO | FP16 quant |
|---|---|---|---|
| **Total Area**($um^2$) | 429,342 | 439,762 | 882,233 |
| **Latency(us)** | 1129.2775 | 1129.2825 | 1129.3150 |

MAC's area and power. The latency results are similar across all three methods because all data oper-

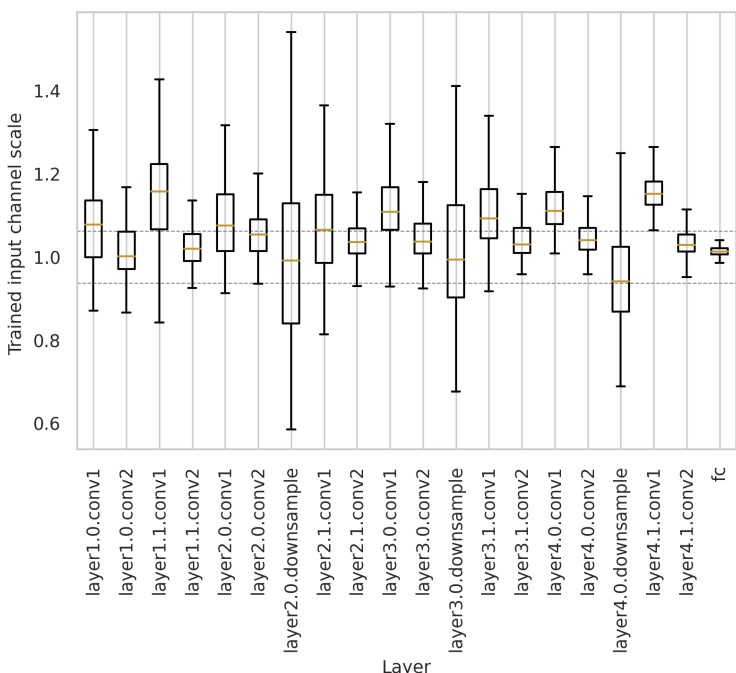

Figure 11: Input channel scale values per layer in ResNet-18 when trained without group constraints.

ates in a pipeline. The speed is comparable if the number of MACs is the same. The additional logic slightly alters the pipeline length, causing a minor increase in latency, but it's negligible compared to the overall data.

# F  CALCULATION COST

## F.1  INTEGER OPERATIONS

Tables 12 and 13 illustrate the computation cost calculation process, as discussed in Chapter 4.4, using ResNet-18 and MobileNetV2 as examples. Initially, each layer is categorized as either Convolution, Fully-connected, or Depthwise-convolution. The number of output features in each layer can be computed as $output channel \times width \times height$. The computational effort required to derive the value of a single output channel is an integer multiplication of the product of the input channel and the width $fx$ and height $fy$ of the kernel, along with an integer addition of $ic \times fx \times fy - 1$ needed for accumulation.

Therefore, the number of integer operations required before applying ICG can be calculated as $(number\ of\ mul + number\ of\ sum) \times (number\ of\ output)$.

Each output feature requiring ICG will require additional multiplications and additions by the number of group values. In the case of Chapter 4.4, the number of groups is 3, and two of these groups have a group scale value that is not equal to 1. Consequently, four multiplications and additions are required for a single output feature.

In a depth-wise convolution layer, input channel grouping is not necessary. Therefore, the calculation cost for the grouping operation is not included in the overall computations.

Table 12: Number of integer operations with ICG in ResNet-18

| | Output Feature | | | Weight | | | DW | | Conv/FC | | | ICG | Total | |
|---|---|---|---|---|---|---|---|---|---|---|---|---|---|---|
| layer | oc | w | h | ic | fx | fy | | # of output | # of mul | # of sum | # of Op. | # of ICG Op. | Total # of Op. | ICG ratio |
| 1 | 64 | 56 | 56 | 64 | 3 | 3 | F | 200,704 | 576 | 575 | 231,010,304 | 802,816 | 231,813,120 | 0.35% |
| 2 | 64 | 56 | 56 | 64 | 3 | 3 | F | 200,704 | 576 | 575 | 231,010,304 | 802,816 | 231,813,120 | 0.35% |
| 3 | 128 | 28 | 28 | 64 | 3 | 3 | F | 100,352 | 576 | 575 | 115,505,152 | 401,408 | 115,906,560 | 0.35% |
| 4 | 128 | 28 | 28 | 128 | 3 | 3 | F | 100,352 | 1,152 | 1,151 | 231,110,656 | 401,408 | 231,512,064 | 0.17% |
| 5 | 256 | 14 | 14 | 128 | 3 | 3 | F | 50,176 | 1,152 | 1,151 | 115,555,328 | 200,704 | 115,756,032 | 0.17% |
| 6 | 256 | 14 | 14 | 256 | 3 | 3 | F | 50,176 | 2,304 | 2,303 | 231,160,832 | 200,704 | 231,361,536 | 0.09% |
| 7 | 512 | 7 | 7 | 256 | 3 | 3 | F | 25,088 | 2,304 | 2,303 | 115,580,416 | 100,352 | 115,680,768 | 0.09% |
| 8 | 512 | 7 | 7 | 512 | 3 | 3 | F | 25,088 | 4,608 | 4,607 | 231,185,920 | 100,352 | 231,286,272 | 0.04% |
| 9 | 64 | 56 | 56 | 64 | 3 | 3 | F | 200,704 | 576 | 575 | 231,010,304 | 802,816 | 231,813,120 | 0.35% |
| 10 | 64 | 56 | 56 | 64 | 3 | 3 | F | 200,704 | 576 | 575 | 231,010,304 | 802,816 | 231,813,120 | 0.35% |
| 11 | 128 | 28 | 28 | 64 | 3 | 3 | F | 100,352 | 576 | 575 | 115,505,152 | 401,408 | 115,906,560 | 0.35% |
| 12 | 128 | 28 | 28 | 128 | 3 | 3 | F | 100,352 | 1,152 | 1,151 | 231,110,656 | 401,408 | 231,512,064 | 0.17% |
| 13 | 256 | 14 | 14 | 128 | 3 | 3 | F | 50,176 | 1,152 | 1,151 | 115,555,328 | 200,704 | 115,756,032 | 0.17% |
| 14 | 256 | 14 | 14 | 256 | 3 | 3 | F | 50,176 | 2,304 | 2,303 | 231,160,832 | 200,704 | 231,361,536 | 0.09% |
| 15 | 512 | 7 | 7 | 256 | 3 | 3 | F | 25,088 | 2,304 | 2,303 | 115,580,416 | 100,352 | 115,680,768 | 0.09% |
| 16 | 512 | 7 | 7 | 512 | 3 | 3 | F | 25,088 | 4,608 | 4,607 | 231,185,920 | 100,352 | 231,286,272 | 0.04% |
| 17 | 1000 | 1 | 1 | 512 | 1 | 1 | F | 1,000 | 512 | 511 | 1,023,000 | 4,000 | 1,027,000 | 0.39% |
| Total | | | | | | | | | | | 3,005,260,824 | 6,025,120 | 3,011,285,944 | 0.20% |

Table 13: Number of integer operations with ICG in mobileNetV2

| | Output Feature | | | Weight | | | DW | | Conv/FC | | | ICG | Total | |
|---|---|---|---|---|---|---|---|---|---|---|---|---|---|---|
| layer | oc | w | h | ic | fx | fy | | # of output | # of mul | # of sum | # of Op. | # of ICG Op. | Total # of Op. | ICG ratio |
| 1 | 32 | 112 | 112 | 1 | 3 | 3 | T | 401,408 | 9 | 8 | 6,823,936 | 0 | 6,823,936 | 0.00% |
| 2 | 96 | 112 | 112 | 16 | 1 | 1 | F | 1,204,224 | 16 | 15 | 37,330,944 | 4,816,896 | 42,147,840 | 12.90% |
| 3 | 96 | 56 | 56 | 1 | 3 | 3 | T | 301,056 | 9 | 8 | 5,117,952 | 0 | 5,117,952 | 0.00% |
| 4 | 144 | 56 | 56 | 24 | 1 | 1 | F | 451,584 | 24 | 23 | 21,224,448 | 1,806,336 | 23,030,784 | 8.51% |
| 5 | 144 | 56 | 56 | 1 | 3 | 3 | T | 451,584 | 9 | 8 | 7,676,928 | 0 | 7,676,928 | 0.00% |
| 6 | 144 | 56 | 56 | 24 | 1 | 1 | F | 451,584 | 24 | 23 | 21,224,448 | 1,806,336 | 23,030,784 | 8.51% |
| 7 | 144 | 28 | 28 | 1 | 3 | 3 | T | 112,896 | 9 | 8 | 1,919,232 | 0 | 1,919,232 | 0.00% |
| 8 | 192 | 28 | 28 | 32 | 1 | 1 | F | 150,528 | 32 | 31 | 9,483,264 | 602,112 | 10,085,376 | 6.35% |
| 9 | 192 | 28 | 28 | 1 | 3 | 3 | T | 150,528 | 9 | 8 | 2,558,976 | 0 | 2,558,976 | 0.00% |
| 10 | 192 | 28 | 28 | 32 | 1 | 1 | F | 150,528 | 32 | 31 | 9,483,264 | 602,112 | 10,085,376 | 6.35% |
| 11 | 192 | 28 | 28 | 1 | 3 | 3 | T | 150,528 | 9 | 8 | 2,558,976 | 0 | 2,558,976 | 0.00% |
| 12 | 192 | 28 | 28 | 32 | 1 | 1 | F | 150,528 | 32 | 31 | 9,483,264 | 602,112 | 10,085,376 | 6.35% |
| 13 | 192 | 14 | 14 | 1 | 3 | 3 | T | 37,632 | 9 | 8 | 639,744 | 0 | 639,744 | 0.00% |
| 14 | 384 | 14 | 14 | 64 | 1 | 1 | F | 75,264 | 64 | 63 | 9,558,528 | 301,056 | 9,859,584 | 3.15% |
| 15 | 384 | 14 | 14 | 1 | 3 | 3 | T | 75,264 | 9 | 8 | 1,279,488 | 0 | 1,279,488 | 0.00% |
| 16 | 384 | 14 | 14 | 64 | 1 | 1 | F | 75,264 | 64 | 63 | 9,558,528 | 301,056 | 9,859,584 | 3.15% |
| 17 | 384 | 14 | 14 | 1 | 3 | 3 | T | 75,264 | 9 | 8 | 1,279,488 | 0 | 1,279,488 | 0.00% |
| 18 | 384 | 14 | 14 | 64 | 1 | 1 | F | 75,264 | 64 | 63 | 9,558,528 | 301,056 | 9,859,584 | 3.15% |
| 19 | 384 | 14 | 14 | 1 | 3 | 3 | T | 75,264 | 9 | 8 | 1,279,488 | 0 | 1,279,488 | 0.00% |
| 20 | 384 | 14 | 14 | 64 | 1 | 1 | F | 75,264 | 64 | 63 | 9,558,528 | 301,056 | 9,859,584 | 3.15% |
| 21 | 384 | 14 | 14 | 1 | 3 | 3 | T | 75,264 | 9 | 8 | 1,279,488 | 0 | 1,279,488 | 0.00% |
| 22 | 576 | 14 | 14 | 96 | 1 | 1 | F | 112,896 | 96 | 95 | 21,563,136 | 451,584 | 22,014,720 | 2.09% |
| 23 | 576 | 14 | 14 | 1 | 3 | 3 | T | 112,896 | 9 | 8 | 1,919,232 | 0 | 1,919,232 | 0.00% |
| 24 | 576 | 14 | 14 | 96 | 1 | 1 | F | 112,896 | 96 | 95 | 21,563,136 | 451,584 | 22,014,720 | 2.09% |
| 25 | 576 | 14 | 14 | 1 | 3 | 3 | T | 112,896 | 9 | 8 | 1,919,232 | 0 | 1,919,232 | 0.00% |
| 26 | 576 | 14 | 14 | 96 | 1 | 1 | F | 112,896 | 96 | 95 | 21,563,136 | 451,584 | 22,014,720 | 2.09% |
| 27 | 576 | 7 | 7 | 1 | 3 | 3 | T | 28,224 | 9 | 8 | 479,808 | 0 | 479,808 | 0.00% |
| 28 | 960 | 7 | 7 | 160 | 1 | 1 | F | 47,040 | 160 | 159 | 15,005,760 | 188,160 | 15,193,920 | 1.25% |
| 29 | 960 | 7 | 7 | 1 | 3 | 3 | T | 47,040 | 9 | 8 | 799,680 | 0 | 799,680 | 0.00% |
| 30 | 960 | 7 | 7 | 160 | 1 | 1 | F | 47,040 | 160 | 159 | 15,005,760 | 188,160 | 15,193,920 | 1.25% |
| 31 | 960 | 7 | 7 | 1 | 3 | 3 | T | 47,040 | 9 | 8 | 799,680 | 0 | 799,680 | 0.00% |
| 32 | 960 | 7 | 7 | 160 | 1 | 1 | F | 47,040 | 160 | 159 | 15,005,760 | 188,160 | 15,193,920 | 1.25% |
| 33 | 960 | 7 | 7 | 1 | 3 | 3 | T | 47,040 | 9 | 8 | 799,680 | 0 | 799,680 | 0.00% |
| 34 | 1280 | 7 | 7 | 320 | 1 | 1 | F | 62,720 | 320 | 319 | 40,078,080 | 250,880 | 40,328,960 | 0.63% |
| 35 | 1000 | 1 | 1 | 1280 | 1 | 1 | F | 1,000 | 1,280 | 1,279 | 2,559,000 | 4,000 | 2,563,000 | 0.16% |
| Total | | | | | | | | | | | 337,938,520 | 13,614,240 | 351,552,760 | 4.03% |

