# OpenReview forum: "Hardware-Friendly Post-Training Quantization: Input- and Output-Channelwise Scale and Offset"
_ICLR.cc/2024/Conference — Submitted to ICLR 2024_

### Official Review · Reviewer_LRrq · 2023-10-31

**Soundness:** 3 good
**Presentation:** 2 fair
**Contribution:** 2 fair
**Rating:** 5
**Confidence:** 5

**Summary:**

Observing the distributional discrepancy between the full precision weights and their quantized counterpart, this paper proposes to scale and offset input and output in a per-channel way.  Extensive experiments are conducted to show the effectiveness of their methods, especially in low-bit settings.

**Strengths:**

1. The paper conducts thorough experiments, including analyses of computational complexity and ablation studies, to showcase the effectiveness of their methods. It also provides a detailed comparison with related works.
2. The proposed methods exhibit significant improvements over previous approaches, such as BRECQ[1], especially in low-bit scenarios.

[1]Yuhang Li, et al. BRECQ: Pushing the Limit of Post-Training Quantization by Block Reconstruction.ICLR 2021

**Weaknesses:**

1. The novelty of the paper raises concerns, as the input-channel scale and shift resemble group-wise quantization have been studied in prior works like Q-BERT. Although the authors emphasize advantages in hardware implementation, the improvement over group-wise quantization appears minor. Can you provide the comparison between group-wise quantization (with or without the power of two scales) in computation complexity and performance?

2. Similar to Weakness.1, shifting and scaling the output per channel is similar to finetune/update the BatchNorm statistics after the quantized convolutions. Does this method still work if we finetune BN after quantization on Conv-BN networks ?(this method is used widely to recover accuracy after quantization)

[2]Sheng Shen,et al. Q-BERT: Hessian Based Ultra Low Precision Quantization of BERT. AAAI 2020

**Questions:**

Is the shift operation expected to introduce more latency than a single integer operation due to non-local memory access at inference?

---

> ### Author Response · Authors · 2023-11-13
> **Response to Reviewer LRrq**
>
> Thank you for taking the time to review our paper thoroughly and for providing valuable comments and feedback. In response to your questions, below are detailed answers to each one. We hope you find them helpful.
>
> **Q1. Compare with other input channel-wise grouping**
>
> **A1-1. Differences from Existing Channel Grouping Methods.**
>
> > Previous works in group-wise quantization, like Q-BERT[12], require each group to have different scale values. As highlighted in QLoRA[9], this requirement creates significant memory overhead during low-bit quantization. For example, if weights for 64 groups are quantized with a single scale per group, it demands 32-bit scales for each of those 64 weights. This scenario is similar to each weight incurring an additional memory overhead of 0.5 bits. While this impact is less pronounced in higher-bit quantizations, in the case of 2-bit quantization, it equates to an approximate 25% increase in memory requirements.
>
> > Our method, instead of creating groups based on unrelated weight positions, undergoes a calibration stage to find the optimal group for predetermined scale values. Since shuffled channels are independent, they can be reorganized through permutation, eliminating the need for additional scale information memory. Furthermore, we opted for shift operations to stay within the integer domain to avoid requiring floating point multipliers and adders if the scale multiplication remains in the floating domain. Table 4 in our ablation study demonstrates that significant performance improvements can be achieved through shift operations alone.
>
> **A1-2. ASIC/FPGA Area/Latency comparison**
>
> > We conducted experiments using the structure of a batch-norm fused systolic array from an NPU developed for edge devices to compare overhead and latency in ASICs. The results below show the area of the batch-norm fused systolic array when each method is implemented in ASIC. The synthesis was done using the TSMC 12nm process. The MAC has a 16x16 structure, and IOSO, batch norm, and activation can process 8 data in 1 cycle. The sequence is MAC(->IOSO)-> BN -> ACT. Please refer to the "Common Response from Authors." Due to time and resource constraints, the actual performance time, or latency, was tested only on the resnet 50 layer with 28x28x128 feature 3x3x128x128 kernel layer.
>
> |  | **integer Quant.** | **+IOSO** | **FP16 quant** |
> |---|---:|---:|---:|
> | **Total Area(um^2)** | 429,342 | 439,762 | 882,233 |
> | **Latency(us)** | 1129.2775 | 1129.2825 | 1129.3150 |
>
> > As seen in the table above, IOSO can be implemented with an integer shifter adder without modifying the MAC structure, resulting in only a 2.37% increase in area. In addition, the implementation is very simple as it does not alter the existing data path. In contrast, methods partially implemented with FP16 points (like GPTQ[10], QLoRA[9], LLM.int8()[11]) for increased accuracy and reduced memory are useful in GPUs that already have FP16 point kernels. However, from an ASIC perspective, these are challenging to use due to significant increases in MAC's area and power. The latency results are similar across all three methods because all data operates in a pipeline. The speed is comparable if the number of MACs is the same. The additional logic slightly alters the pipeline length, causing a minor increase in latency, but it's negligible compared to the overall data.
>
> **Q2. Output channel-wise scaling/offset.**
> > We included output channel-wise adjustments because they showed better results when learned together with rounding and input channel scaling. Since input channel-wise offset is not adjustable, any bias that may arise from input channel-wise scaling is corrected through an offset in the output channel. As can be seen in Table 4 of our ablation study, better results were achieved when all elements were adjusted together rather than learning each item separately. This approach ensures that the combined effect of these adjustments leads to more effective and balanced performance improvements.
>
> **Q3. Shift operation overhead.**
> > We can consider the scenarios for GPUs and ASICs separately. In the case of GPUs, shift operations need to be performed on accumulated values to be processed similarly to adders(accumulators). For ASICs, there are two possibilities. When using a fixed $\gamma_y$, the operation can be implemented through wire connections, eliminating timing issues. If the number of shifts in gamma_y is variable, it can be implemented with a multiplexer (mux) for each required shift. In this context, as demonstrated in Experiment Figure 6, values of N up to 6 are meaningful, which means only a small number of multiplexers are needed.
>
> **References.**
>
> Please refer to the "Common Response from Authors" mentioned above.

---

> > ### Comment · Reviewer_LRrq · 2023-11-22
> > **Reply to the authors' responses**
> >
> > Thank you for the responses. Part of my concerns are solved.
> >
> > Is it because the proposed method use less scale values, thus the memory overhead during low-bit quantization can be reduced? If it was the case, could you provide some further comments about the differences with RPTQ?
> >
> > The authors only explained the motivation of channel-wise scaleing/offset. But the difference between channel-wise scaling/offset and finetuning BatchNormalization is not discussed. It seems to me that these two methods are the same.

---

> ### Author Response · Authors · 2023-11-22
> **Response to Reviewer LRrq**
>
> I would like to express my gratitude to the reviewer for his/her response and valuable constructive feedback. To address the queries raised, I have provided more detailed responses in the section below.
>
> **Q1. Is it because the proposed method use less scale values, thus the memory overhead during low-bit quantization can be reduced? If it was the case, could you provide some further comments about the differences with RPTQ?**
>
> Rather than saying memory is reduced due to the use of a small number of scale values, it would be more accurate to say that memory usage does not increase because there are predefined scale values for the model.
>
> The comparison with RPTQ is as follows.
>
> * **Similarities**:
> 	+ Both use memory reorder to cluster channels with similar characteristics.
> 	+ Reordering allows efficient use of batch matrix multipliers (bmm).
>
> * **Differences**:
> 	* Memory usage of scale value:
> 		* **RPTQ**: Each cluster of each layer stores a separate 32-bit floating scale value. More clusters require additional storage for scale values.
> 		* **IOSO**: The model has three predetermined $\gamma^G$ values. Each layer does not need its scale value; only one 3-bit integer (value between 1-7) is required. There are three clusters.
> 	* Optimization method:
> 		* **RPTQ**: Uses K-Means algorithm for clustering activation min/max.
> 		* **IOSO**: Finds groups that minimize reconstruction error.
> 	* Application of scale value:
>
> 		* **RPTQ**: On activation quantization.
> 		* **IOSO**: On weight and activation product of quantized value.
> 	* Optimization speed:
> 		* **RPTQ**: Very fast. It only requires the K-Means algorithm with a 2-dimensional space ($X_{min}$, $X_{max}$).
> 		* **IOSO**: Relatively slower as it needs to calculate the loss. However, it is almost similar to the round-based methods's speed as it involves additional learning parameters.
> In summary, RPTQ is specialized for speed for application in Large Language Models, like other LLM quantization methods. Therefore, it focuses on reducing quantization error through straightforward techniques (min/max quantization, K-means). Our IOSO, although slower than LLM quantization methods, shows lower reconstruction error (quantization error).
>
> Below are the experimental results to quantitatively compare the two methods. For quick results, please understand that we only tested the first block of resnet-18 (.model.layer1.0.conv1, .conv2). We used the same 3 clusters and only a 1024 calibration set.
> | Method | BASE | IOSO | IOSO | IOSO | RPTQ |
> |---|---|---|---|---|---|
> | **cycle** |  | 1000 | 2500 | 625 | - |
> | **Time** |  | 46s | 11s | 2s | 52.4ms ± 277 µs  |
> | **Recon Loss** | 126.52 | 57.39 | 59.05 | 108.75 | 98.3 |

---

> ### Author Response · Authors · 2023-11-23
> **Response to Reviewer LRrq**
>
> **Q2. The authors only explained the motivation of channel-wise scaling/offset. But the difference between channel-wise scaling/offset and finetuning BatchNormalization is not discussed. It seems to me that these two methods are the same.**
>
> Methodologically, I agree with the reviewer's concern that output channel-wise scaling/offset can be achieved through batch-norm fine-tuning. Our *output-channel-wise scaling/offset* essentially adopts the parameters used in batch normalization and applies input scaling and rounding error-aware fine-tuning. And from the perspective of applying the method, we have presented the following perspectives:
>
> **1. [With] When used in conjunction with Input Channel-wise scaling and rounding-based quantization, better results can be obtained (Chapters 3.2 and 4.2).**
>
> > Input Channel scaling alone was insufficient to prevent bias from the input channel.
> From Equation (6), it becomes
> $z_k= \gamma^z_k \alpha_k\sum^g_p\gamma^G_p\sum_{i \in G_p}w_{ki}x_{i}+\alpha_k\sum_{i }^{c}\varphi^y_i +\gamma_k^z\beta_k+\varphi^z_k$.
> The presence of the $\alpha_k\sum_{i }^{c}\varphi^y_i$(*input channel bias*) necessitates additional floating addition operations.
> When only Input Channel scaling is applied, it cannot prevent bias arising from the input channel/rounding. Therefore, to reduce this, we incorporate the role in output-wise offset and scaling to mitigate these effects.
>
> **2. [How] Applied to reduce *Block-Based* reconstruction error (Chapter 3.4).**
> >While focusing on batch-norm finetuning could be effective layer-wisely and applicable in reducing the final loss, we found the scale and offset that further reduce loss at the block level. For instance, when applying rounding/input channel-wise scaling at the block level and output channel scaling/offset at the layer level to reduce reconstruction error, there was a performance decrease of 66.62 (-0.60%) in resnet18 at 2/32 bit.
>
> **3. [Where] The placement of output scaling/offset (before/after activation) - (Appendix D.7).**
> > It's possible to apply scaling/offset after activation, not necessarily at the current batch-norm position, but we found that applying it at the conventional batch-norm position generally yields slightly better performance.
>
> **4. [Extension] Additionally, in models like transformers that do not use batch normalization, the performance can be enhanced with just 1 add/mult floating operation per activation (Chapter 3.4).**

---

### Official Review · Reviewer_Zj2N · 2023-11-01

**Soundness:** 3 good
**Presentation:** 3 good
**Contribution:** 2 fair
**Rating:** 6
**Confidence:** 5

**Summary:**

This paper extends AdaRound quantization scheme and introduces a group-based scaling factor along input channel direction. A output-channel-wise, learnable scale and offset are also applied to better reconstruct the low bit cases. To lower the computational cost, author proposed to use bit-wise shifter and only allow scaling factors of 1+- 2^-N. Considering the hardware efficiency of implementing such input channel-wise methods, the author demonstrated the feasibility of setting contraints on the number of the channels per group (to be greater than a certain value) then use retraining to recover the accuracy. Extensive experiments on different CV models and comparisons with other similar PTQ schemes, such as AdaRound, AdaQuant, BRECQ, and QDROP, were provided.

**Strengths:**

1. Clear and detailed explanations about the numerical method.

2. Extensive experiments results, especially the author provides statistics, i.e. 5-run mean and standard deviation, instead of best records. This will help readers to get a better idea while comparing the proposed method with different quantization schemes.

**Weaknesses:**

1. Lack of real HW results. It's clear that the author thoroughly considered the potential limitations if the proposed method was to be implemented. However, there are still some potential concerns, such as channel permutation's impact on computation efficiency and grouping fragments effect. A few examples on a representative HW would make the paper much stronger and convincing.

2. Marginal improvement compared to previous works. It is understandable that the existing methods may have already done decent jobs and the author has to use extreme low precision settings (W2A4) to demonstrate the benefit of the proposed method. However, considering the complexity of implementation and the uncertainty in compute efficiency trade-off, the author might need to find a few better examples where the use of the proposed method would be better justified.

**Questions:**

1. Eq.9 and the explanations in the following paragraph shows that the group scaling factor, gamma_y, is a linear combination of preset gamm_G based on probability. However, that will likely make gamma_y not compatible with bit shifter. In AdaRound, the handling of h(V) is different during calibration/PTQ stage and inference stage. Author might want to clarify/comment on this part or it might cause confusion to the readers.

2. Based on the example in Fig. 6, the 3 scaling factors used are 1 and 1+-2^-N. One may interpret this scheme as some input channels will be up-weighted, some will be down-weighted, and the remaining channels will be unchanged. However, the optimized factor here seems to be very small, implying that simply not applying the scaling factor might work as well? But comparing to Table 4, it seems like the case without input scaling will be close to the case of N=1? Please comment.

3. Instead of using unstructured input channel-wise grouping, another frequently use quantization scheme is structured grouping, such as used by GPTQ and other LLM works. Maybe the author could consider including a few comments on the pros and cons with the proposed method?

---

> ### Author Response · Authors · 2023-11-13
> **Response to Reviewer Zj2N on weaknesses 1**
>
> Thank you for taking the time to review our paper thoroughly and for providing valuable comments and feedback. In response to your questions, below are detailed answers to each one. We hope you find them helpful.
>
> **W1. Real HW results**
>
> **A1-1. Our Objective**
>
> > For hardware, especially ASICs/FPGAs, to operate with low power and low area, the most computation-intensive Multiply-Accumulate (MAC) operations must be performed using integer calculations. To achieve this, weights and activations must be quantized into the smallest possible bit integers. We believe our method, compared to previously studied round-only based Post-Training Quantization (PTQ), offers an option for better accuracy with no additional cost for Output channel-wise (utilizing batch-norm) quantization and low hardware cost for Input Channel-wise operations, enhancing expressiveness.
>
> **A1-2. ASIC/FPGA Area/Latency comparison**
>
> > We conducted experiments using the structure of a batch-norm fused systolic array from an NPU developed for edge devices to compare overhead and latency in ASICs. The results below show the area of the batch-norm fused systolic array when each method is implemented in ASIC. The synthesis was done using the TSMC 12nm process. The MAC has a 16x16 structure, and IOSO, batch norm, and activation can process 8 data in 1 cycle. The sequence is MAC(->IOSO)-> BN -> ACT. For more detailed information, please refer to the "Common Response from Authors." Due to time and resource constraints, the actual performance time, or latency, was tested only on the resnet 50 layer with 28x28x128 feature 3x3x128x128 kernel layer.
>
> |  | **integer Quant.** | **+IOSO** | **FP16 quant** |
> |---|---:|---:|---:|
> | **Total Area(um^2)** | 429,342 | 439,762 | 882,233 |
> | **Latency(us)** | 1129.2775 | 1129.2825 | 1129.3150 |
>
> > As seen in the table above, IOSO can be implemented with an integer shifter adder without modifying the MAC structure, resulting in only a 2.37% increase in area. In addition, the implementation is very simple as it does not alter the existing data path. In contrast, methods partially implemented with FP16 points (like GPTQ[10], QLoRA[9], LLM.int8()[11]) for increased accuracy and reduced memory are useful in GPUs that already have FP16 point kernels. However, from an ASIC perspective, these are challenging to use due to significant increases in MAC's area and power. The latency results are similar across all three methods because all data operates in a pipeline. The speed is comparable if the number of MACs is the same. The additional logic slightly alters the pipeline length, causing a minor increase in latency, but it's negligible compared to the overall data.
>
> **A1-3. IOSO overhead on GPU**
>
> |  | **Baseline(ms)** | **IOSO(ms)** | **Increased latency** |
> |:---|---:|---:|---:|
> | **resnet18** | 15.37±1.37 | 17.13±1.66 | 111.5% |
> | **resnet50** | 38.67±0.83 | 42.26±1.58 | 108.1% |
> | **mobilenet_v2** | 29.94±2.74 | 34.53±2.87 | 117.3% |
> | **regnetx_600m** | 36.29±3.65 | 40.11±1.77 | 110.8% |
>
> > We indirectly tested the latency using a non-optimized (naive) GPU. After adding the add and shift operations to the output feature in the baseline and running the ImageNet validation set with a batch size of 64, we observed the latency results. We used the RTX3080 GPU. On the non-optimized GPU, there was about a max of 17% increase in latency. We expect this value to represent the upper bound of our method's latency.  The latency could further decrease if there is support for a CUDA kernel for IOSO or operations after the accumulator, such as shift-add.

---

> > ### Comment · Reviewer_Zj2N · 2023-11-21
> >
> > With the inclusion of ASIC and GPU HW experiment results, I have adjusted my review rating from 5 to 6. Thanks!

---

> > > ### Author Response · Authors · 2023-11-23
> > > **Thank you for acknowledging our contribution.**
> > >
> > > Thank you for acknowledging our contribution. We will include the additional results in the appendix.

---

> ### Author Response · Authors · 2023-11-13
> **Response to Reviewer Zj2N on weaknesses 2 and questions 1-3**
>
> **W2.  The benefit of the proposed methods.**
>
> **A2-1. Orthogonally applied to round-only based PTQ**
>
> > Our method can be used as an additional option to enhance performance with low hardware cost when low-bit accuracy is insufficient in various existing round-only-based PTQ methods. During the round-only based PTQ process, which is one of the promising directions in PTQ research (e.g., Adaround[1], BRECQ[2], QDROP[3], NWQ[4]), our method can increase the representational power of the quantized network through grouping and offset/scaling. As shown in **A1-2**, our method is simple to implement and can enhance performance with a modest 2.37% increase in area. In contrast, alternative methods either require floating-point operations (e.g., QloRA[9], LLM.int8()[11]) or additional memory for group operations (e.g., GPTQ[10]). Appendix Tables 6 and 7 demonstrate that our method can be orthogonally applied to round-only based PTQ, yielding better performance in most cases.
>
> **A2-2. Ease of Implementation**
>
> > Our approach can be easily implemented by adding grouping and offset/scaling parameters while learning rounding in existing round-based Post-Training Quantization (PTQ) processes. Additionally, as described in **A1-2**, it can be easily incorporated into the existing datapath from a hardware perspective.
>
> **Q3.  Clarify group scaling factor.**
> > The value of $\gamma_y$, as depicted in Figure 3, is predetermined as a hyper-parameter (we used values $1.0$, $1.0-2^{−4}$, and $1.0+2^{−4}$) and is determined by the calibration/PTQ stage ratio. However, these values tend to converge towards one side during calibration, allowing them to function as shift operations at inference. The variable corresponding to $h(V)$ is $s(R_{ij})$. I will offer a more comprehensive explanation of the concept of $s(R_{ij})$ to enhance clarity in the rebuttal.
>
> **Q4. Scaling factor applying**
> > As commented by the reviewer, some channels increase, others decrease, and the rest remain unchanged. However, when $N=\infty$, channel scaling has no effect, similar to not applying any scaling at all, whereas at $N=1$, it has the most significant impact (50%, 100%, 150% scaling). As observed in Fig 6., excessive scaling ($N=1$) degrades performance, while moderate Group Gamma $N$ ($N=4$ for Weight Only, $N=6$ for Weight+Feature) yields the best results, outperforming very small scaling ($N=7$). Increasing $N$ gradually results in outcomes similar to those in Table 4's output scaling/offset case.
>
> **Q5. Compare with other input channel-wise grouping**
> > Previous works on group-wise quantization require each group to maintain different scale values, which, as indicated in QLoRA[9], become a significant memory overhead during low-bit quantization. For instance, quantizing weights for 64 groups with a single scale per group necessitates 32-bit scales for each of the 64 weights. This is akin to each weight carrying an overhead of 0.5 bits of memory. While the impact is less significant in larger bit quantizations, it becomes comparable to a 25% memory increase for 2-bit quantization.
>
> > Instead of creating groups based on unrelated weight positions, our method undergoes a calibration stage to find the optimal group for predetermined scale values. Since shuffled channels are independent, they can be reorganized through permutation, eliminating the need for additional scale information memory. Furthermore, we opted for shift operations to stay within the integer domain to avoid requiring floating point multipliers and adders if the scale multiplication remains in the floating domain. Table 4 in our ablation study demonstrates that significant performance improvements can be achieved through shift operations alone.
>
> > Similar to what was mentioned in A1-2, methods like GPTQ and LLM.int8() keep part or all of the activations in a floating point to reduce memory usage. While this is an effective approach for GPUs, where power and area are not major constraints, it poses a problem for edge devices requiring low power and small-area implementation. This distinction highlights the importance of considering the specific hardware environment when choosing a quantization strategy, as techniques suitable for high-resource settings may not be feasible or efficient in more constrained environments like edge devices.
>
> **References.**
>
> Please refer to the "Common Response from Authors" mentioned above.

---

### Official Review · Reviewer_J41k · 2023-11-02

**Soundness:** 2 fair
**Presentation:** 3 good
**Contribution:** 2 fair
**Rating:** 5
**Confidence:** 4

**Summary:**

This paper presents a quantization method that exploits channel-wise scaling and offset parameters to compensate for the discrepancies in full-precision and quantized distributions. It has been shown that the proposed method outperforms existing works in terms of accuracy.

**Strengths:**

1) The proposed method is simple and effective in quantizing convolutional networks especially when extreme quantization levels are used.

2) The paper is easy to read and understand.

**Weaknesses:**

1) There is no discussion on why the proposed method is considered hardware-friendly. Which hardware is this work referring to? How its efficiency was measured?

2) The number of operations and parameters for each quantization method must be compared along with the accuracy in Table 2 and 3. The accuracy improvement of this work is marginal for most cases. So, it's important to compare other aspects too.

**Questions:**

See the questions listed as weaknesses.

---

> ### Author Response · Authors · 2023-11-13
> **Response to Reviewer J41k on question 1**
>
> Thank you for taking the time to review our paper thoroughly and for providing valuable comments and feedback. In response to your questions, below are detailed answers to each one. We hope you find them helpful.
>
> **Q1. Why is it  Hardware-friendly?**
>
> **A1-1. Our Objective**
>
> > For hardware, especially ASICs/FPGAs, to operate with low power and low area, the most computation-intensive Multiply-Accumulate (MAC) operations must be performed using integer calculations. To achieve this, weights and activations must be quantized into the smallest possible bit integers. We believe our method, compared to previously studied round-only based Post-Training Quantization (PTQ), offers an option for better accuracy with no additional cost for Output channel-wise (utilizing batch-norm) quantization and low hardware cost for Input Channel-wise operations, enhancing expressiveness.
>
> **A1-2. ASIC/FPGA Area/Latency comparison**
>
> > We conducted experiments using the structure of a batch-norm fused systolic array from an NPU developed for edge devices to compare overhead and latency in ASICs. The results below show the area of the batch-norm fused systolic array when each method is implemented in ASIC. The synthesis was done using the TSMC 12nm process. The MAC has a 16x16 structure, and IOSO, batch norm, and activation can process 8 data in 1 cycle. The sequence is MAC(->IOSO)-> BN -> ACT. For more detailed information, please refer to the "Common Response from Authors." Due to time and resource constraints, the actual performance time, or latency, was tested only on the resnet 50 layer with 28x28x128 feature 3x3x128x128 kernel layer.
>
> |  | **integer Quant.** | **+IOSO** | **FP16 quant** |
> |---|---:|---:|---:|
> | **Total Area(um^2)** | 429,342 | 439,762 | 882,233 |
> | **Latency(us)** | 1129.2775 | 1129.2825 | 1129.3150 |
>
> > As seen in the table above, IOSO can be implemented with an integer shifter adder without modifying the MAC structure, resulting in only a **2.37%** increase in area. In addition, the implementation is very simple as it does not alter the existing data path. In contrast, methods partially implemented with FP16 points (like QLoRA[9], GPTQ[10], and LLM.int8()[11]) for increased accuracy and reduced memory are useful in GPUs that already have FP16 point kernels. However, from an ASIC perspective, these are challenging to use due to significant increases in MAC's area and power. The latency results are similar across all three methods because all data operates in a pipeline. The speed is comparable if the number of MACs is the same. The additional logic slightly alters the pipeline length, causing a minor increase in latency, but it's negligible compared to the overall data.
>
> **A1-3. IOSO overhead on GPU**
> |  | **Baseline(ms)** | **IOSO(ms)** | **Increased latency** |
> |:---|---:|---:|---:|
> | **resnet18** | 15.37±1.37 | 17.13±1.66 | 111.5% |
> | **resnet50** | 38.67±0.83 | 42.26±1.58 | 108.1% |
> | **mobilenet_v2** | 29.94±2.74 | 34.53±2.87 | 117.3% |
> | **regnetx_600m** | 36.29±3.65 | 40.11±1.77 | 110.8% |
>
> > We indirectly tested the latency using a non-optimized (naive) GPU. After adding the add and shift operations to the output feature in the baseline and running the ImageNet validation set with a batch size of 64, we observed the latency results. We used the RTX3080 GPU. There was about a max 17% increase in latency on the non-optimized GPU. We expect this value to represent the upper bound of our method's latency. The latency could further decrease if there is support for a CUDA kernel for IOSO or operations after the accumulator, such as shift-add.

---

> ### Author Response · Authors · 2023-11-13
> **Response to Reviewer J41k on question 2**
>
> **Q2.  Compare other aspects.**
>
> **A2-1. Orthogonally applied to round-only based PTQ**
>
> > Our method can be used as an additional option to enhance performance with low hardware cost when low-bit accuracy is insufficient in various existing round-only-based PTQ methods. During the round-only based PTQ process, which is one of the promising directions in PTQ research (e.g., Adaround[1], BRECQ[2], QDROP[3], NWQ[4]), our method can increase the representational power of the quantized network through grouping and offset/scaling. As shown in **A1-2**, our method is simple to implement and can enhance performance with a modest 2.37% increase in area. In contrast, alternative methods either require floating-point operations (e.g., QloRA[9], LLM.int8()[11]) or additional memory for group operations (e.g., GPTQ[10]). Appendix Tables 6 and 7 demonstrate that our method can be orthogonally applied to round-only based PTQ, yielding better performance in most cases.
>
> **A2-2. Differentiation of previous works on group-wise quantization**
>
> > Previous works on group-wise quantization require each group to maintain different scale values, which, as indicated in QLoRA[9], become a significant memory overhead during low-bit quantization. For instance, quantizing weights for 64 groups with a single scale per group necessitates 32-bit scales for each of the 64 weights. This is akin to each weight carrying an overhead of 0.5 bits of memory. While the impact is less significant in larger bit quantizations, it becomes comparable to a 25% memory increase for 2-bit quantization.
>
> > Instead of creating groups based on unrelated weight positions, our method undergoes a calibration stage to find the optimal group for predetermined scale values. Since shuffled channels are independent, they can be reorganized through permutation, eliminating the need for additional scale information memory. Furthermore, we opted for shift operations to stay within the integer domain to avoid requiring floating point multipliers and adders if the scale multiplication remains in the floating domain. Table 4 in our ablation study demonstrates that significant performance improvements can be achieved through shift operations alone.
>
> **References.**
>
> Please refer to the "Common Response from Authors" mentioned above.

---

### Official Review · Reviewer_yDxC · 2023-11-02

**Soundness:** 3 good
**Presentation:** 4 excellent
**Contribution:** 2 fair
**Rating:** 8
**Confidence:** 4

**Summary:**

The paper proposes an improved post training quantization (PTQ) method called IOSO. The method relies on adjusting the scale and offset in activations (both input and output). The benefits are shown for ImageNet on a diverse set of DNNs.

**Strengths:**

1) The paper is well-written and easy to understand.
2) The paper addresses a well-addressed problem and tries to make a contribution.
3) The improvements in accuracy are non-trivial in some scenarios.
4) The computational overhead is small < 1.5%.
5) Error bars are shown for network accuracy.

**Weaknesses:**

1. The results in Table 2 are incremental. This is not surprising since post-training quantization has been studied extensively.
2. The proposed methods minimizes layer-wise reconstruction error. Reconstruction error is a proxy for misclassification error, which is the ultimate metric. Comparison with methods that directly minimize misclassification error is missing.

**Questions:**

1. There are PTQ methods, e.g., [1], that minimize the probability of misclassification. How does your method compare with those?
[1] Sakr et. al. Analytical guarantees on the numerical precision of deep neural networks, ICML'17.
2. Have you considered fine-tuning your results by doing some limited amount of training, say one epoch? It will be interesting to see if the accuracy improves significantly.
3. Can your method be extended to cover training?

---

> ### Author Response · Authors · 2023-11-13
> **Response to Reviewer yDxC**
>
> Thank you for taking the time to review our paper thoroughly and for providing valuable comments and feedback. In response to your questions, below are detailed answers to each one. We hope you find them helpful.
>
> **Q1. How does your method compare to PTQ methods that aim to minimize misclassification?**
>
> **A1-1**. Compared with “Analytical guarantees on the numerical precision of deep neural networks.”
>
> > Compared to the paper "Analytical guarantees on the numerical precision of deep neural networks,"[8] they calculated the fixed-point operation cost from a computational/storage space perspective. They analyzed the theoretical bounds when transitioning from floating point to fixed point operations. While similarly focusing on computational and representation costs, our method differs in its approach. While [8] theoretically specified an upper bound for bits and solved it with fixed quantization, our method learns to minimize each layer's reconstruction loss through a calibration set.
>
> **A-2** Compared with other PTQ methods.
>
> > Compared with other PTQ methods, approaches like GPTQ, ZeroQuant, LLM.int8(), and QLoRA have been recently proposed.
> We aim to enable integer operations of MAC calculations, the major computational element, for low power/area efficiency in hardware, especially ASIC/FPGA. This requires quantizing weights and activations into as few bits of integers as possible.
> Traditional methods only quantized the weight, necessitating MAC operations in FP32/16 (as in GPTQ[10], QLoRA[11]),  partially retained FP16 (like in LLM.int8()[9]) or demanding additional memory for Group Quantization. These methods significantly increased power and area costs in ASIC/FPGA hardware developed mainly for GPU memory reduction without accuracy loss.
> Our method differs as it improves the performance of traditional round-only based methods (such as AdaRound, BRECQ[1], and QDROP[2]) without significantly increasing the cost in ASIC/FPGA settings.
>
> **Q2. Training amount limit?**
>
> **A2**.
> > Our method involves finding appropriate rounding values while simultaneously performing grouping and scaling/offset. We couldn't achieve favorable results with just one epoch (1024 samples). Also, fixing the rounding and performing grouping later did not yield satisfactory outcomes.
> Additionally, rounding and grouping use a temperature hyperparameter, and the importance of each determines the specific group/rounding. Therefore, a certain number of epochs are necessary. In the presented results, we used 35,000 cycles (about 2187 epochs) and found that at least 15,000 cycles (about 937 epochs) are required to achieve satisfactory results.
>
> **Q3. Conventional training?**
>
> **A3**.
> > Based on my understanding of the reviewer's question, it seems to refer not to post-training quantization but to the use of our method in normal training. We assumed training minimizes changes to the given pre-trained weights within a limited dataset (small calibration set). Therefore, we calibrated the network using rounding and scaling/offset.
> Applying our method to normal training without quantization seems meaningless. If the optimal point requires specific channel scaling/biasing, training the weights in that direction would be more appropriate.
> However, the method might be meaningful in quantization-aware training (QAT) in low-bit quantization. It can enhance the representational power of quantized weights/activation through scaling factors and offsets.
>
> ###  **References.**
>
> Please refer to the "Common Response from Authors" mentioned above.

---

### Author Response · Authors · 2023-11-13
**Common Response from Authors**

### **Implementation Details:**
This is a Systolic Array + Batch_norm + Activation Fused structure of a Neural Processing Unit (NPU) developed for edge devices. The results are synthesized using TSMC’s 12nm 0.9 process, 400Mhz(2.5ns per cycle), and the unit of measurement is square micrometers ($\mu m^2$). The performance of each module is as follows:

* MAC: A 16(input channel) x 16(output channel) MAC array.
* IOSO: Consists of 8 integer adders and 8 integer shifters.
* NORM: Includes 8 floating-point adders and multipliers.
* NORM_LUT: Memory allocated for normalization coefficients.
* Act: Contains 8 activation logic units.
* Quant: Comprises 8 quantization modules.


### **ASIC Synthesis Area Summary**
* Base quantization
|  | **Combi** | **FF** | **Logic(Combi+FF)** | **Mem** | **Total(Logic+Mem)** | **Ratio** |
|:---:|---:|---:|---:|---:|---:|---:|
| **Total** | 211,145 | 99,673 | 310,818 | 118,524 | 429,342 | 100.00% |
| `MAC` | 123,648 | 44,416 | 168,064 | 0 | 168,064 | 39.14% |
| `norm` | 70,844 | 27,507 | 98,351 | 0 | 98,351 | 22.91% |
| `norm_LUT` | 2,676 | 21,013 | 23,689 | 118,524 | 142,213 | 33.12% |
| `Act` | 897 | 1,781 | 2,678 | 0 | 2,678 | 0.62% |
| `Quant` | 11,243 | 2,912 | 14,155 | 0 | 14,155 | 3.30% |
| `ETC` | 1,837 | 2,044 | 3,881 | 0 | 3,881 | 0.90% |

* Base quantization with IOSO
|  | **Combi** | **FF** | **Logic(Combi+FF)** | **Mem** | **Total(Logic+Mem)** | **Ratio** |
|:---:|---:|---:|---:|---:|---:|---:|
| **Total** | 216,814 | 104,424 | 321,238 | 118,524 | 439,762 | 100.00% |
| `MAC` | 123,606 | 44,417 | 168,023 | 0 | 168,023 | 38.21% |
| `IOSO` | 5,646 | 4,768 | 10,414 | 0 | **10,414** | **2.37%** |
| `norm` | 70,903 | 27,499 | 98,402 | 0 | 98,402 | 22.38% |
| `norm_LUT` | 2,678 | 21,003 | 23,681 | 118,524 | 142,205 | 32.34% |
| `Act` | 897 | 1,781 | 2,678 | 0 | 2,678 | 0.61% |
| `Quant` | 11,246 | 2,911 | 14,157 | 0 | 14,157 | 3.22% |
| `ETC` | 1,838 | 2,045 | 3,883 | 0 | 3,883 | 0.88% |

* FP16 MAC quantization
|  | **Combi** | **FF** | **Logic(Combi+FF)** | **Mem** | **Total(Logic+Mem)** | **Ratio** |
|:---:|---:|---:|---:|---:|---:|---:|
| **Total** | 664,053 | 99,656 | 763,709 | 118,524 | 882,233 | 100.00% |
| `MAC` | 576,464 | 44,418 | 620,882 | 0 | **620,882** | **70.38%** |
| `norm` | 70,928 | 27,506 | 98,434 | 0 | 98,434 | 11.16% |
| `norm_LUT` | 2,680 | 20,996 | 23,676 | 118,524 | 142,200 | 16.12% |
| `Act` | 897 | 1,780 | 2,677 | 0 | 2,677 | 0.30% |
| `Quant` | 11,245 | 2,913 | 14,158 | 0 | 14,158 | 1.60% |
| `ETC` | 1,839 | 2,043 | 3,882 | 0 | 3,882 | 0.44% |

### **References.**

[1] Markus Nagel, Rana Ali Amjad, Mart Van Baalen, Christos Louizos, Up or Down? Adaptive Rounding for Post-Training Quantization. (ICML 2020)

[2] Yuhang Li, Ruihao Gong, Xu Tan, Yang Yang, Peng Hu, Qi Zhang, Fengwei Yu, Wei Wang, and Shi Gu. Brecq: Pushing the limit of post-training quantization by block reconstruction. (ICLR 2021)

[3] Xiuying Wei, Ruihao Gong, Yuhang Li, Xianglong Liu, and Fengwei Yu. Qdrop: randomly dropping quantization for extremely low-bit post-training quantization. (ICLR 2022)

[4] Jung Hyun Lee, Jeonghoon Kim, Se Jung Kwon, Dongsoo Lee FlexRound: Learnable Rounding based on Element-wise Division for Post-Training Quantization (ICML2023)

[5] Steven K. Esser, Jeffrey L. McKinstry, Deepika Bablani, Rathinakumar Appuswamy, Dharmendra S. Modha .Learned Step Size Quantization (ICLR2020)

[6] Itay Hubara, Yury Nahshan, Yair Hanani, Ron Banner, Daniel Soudry. Improving Post Training Neural Quantization: Layer-wise Calibration and Integer Programming (arXiv:2006.10518 (2020))

[7] Julian Faraone, Nicholas Fraser, Michaela Blott, Philip H.W. Leong .SYQ: Learning Symmetric Quantization for Efficient Deep Neural Networks (CVPR2018)

[8] C Sakr, Y Kim, N Shanbhag. Analytical guarantees on the numerical precision of deep neural networks (ICML2017)

[9] Tim Dettmers, Artidoro Pagnoni, Ari Holtzman, Luke Zettlemoyer. QLoRA: Efficient Finetuning of Quantized LLMs(arXiv:2305.14314)

[10] Elias Frantar, Saleh Ashkboos, Torsten Hoefler, Dan Alistarh. GPTQ: Accurate Post-Training Quantization for Generative Pre-trained Transformers(ICLR2023)

[11] Tim Dettmers, Mike Lewis, Younes Belkada, Luke Zettlemoyer. LLM.int8(): 8-bit Matrix Multiplication for Transformers at Scale(NeurIPS2022)

[12] Shen, S., Dong, Z., Ye, J., Ma, L., Yao, Z., Gholami, A., ... & Keutzer, K. Q-bert: Hessian based ultra low precision quantization of bert. (AAAI2020)

---

### Meta-Review · Area_Chair_bXKn · 2023-12-14

**Metareview:**

The paper was reviewed by 4 experts and received mixed scores. One reviewer was the most positive about the work and provided higher rating. Their review, however, is superficial and doesn't contain enough information and evidence to support such high rating. Other reviewers were less positive, although not entirely negative. They highlighted the overall effectiveness of the method and clear presentation. At the same time, they had concerns such as lack of real HW results, marginal improvements and similarity to previous works in terms of methodology. The authors addressed some of the concerns, but also agreed to an extend with some of the concerns. AC didn't find enough support to accept the manuscript and decided to recommend rejection in this case.

**Justification For Why Not Higher Score:**

The paper can get a higher score, but it's clearly a borderline paper. The negative reviewers are the most convincing, while the most positive reviewer is not entirely certain and provided superficial review. This review, in turned, raised the score of the paper.

**Justification For Why Not Lower Score:**

N/A

---

### Decision · Program_Chairs · 2024-01-16

Reject